# EQUIVARIANT SCORE-BASED GENERATIVE MODELS PROVABLY LEARN DISTRIBUTIONS WITH SYMMETRIES EFFICIENTLY

## ABSTRACT

Symmetry is ubiquitous in many real-world phenomena and tasks, such as physics, images, and molecular simulations. Empirical studies have demonstrated that incorporating symmetries into generative models can provide better generalization and sampling efficiency when the underlying data distribution has group symmetry. In this work, we provide the first theoretical analysis and guarantees of score-based generative models (SGMs) for learning distributions that are invariant with respect to some group symmetry and offer the first quantitative comparison between data augmentation and adding equivariant inductive bias. First, building on recent works on the Wasserstein-1 ($\mathbf{d}_1$) guarantees of SGMs and empirical estimations of probability divergences under group symmetry, we provide an improved $\mathbf{d}_1$ generalization bound when the data distribution is group-invariant. Second, we rigorously demonstrate that one can learn the score of a symmetrized distribution using equivariant vector fields without data augmentations through the analysis of the optimality and equivalence of score-matching objectives. This also provides practical guidance that one does not have to augment the dataset as long as the vector field or the neural network parametrization is equivariant. Then we quantify the impact of not incorporating equivariant structure into the score parametrization, by showing that non-equivariant vector fields can yield worse generalization bounds. This can be viewed as a type of model-form error that describes the missing structure of non-equivariant vector fields. Third, we describe the inductive bias of equivariant SGMs using Hamilton-Jacobi-Bellman theory. Numerical simulations corroborate our analysis and highlight that data augmentations cannot replace the role of equivariant vector fields.

## 1 INTRODUCTION

Improving data efficiency and reducing computational costs are central concerns in generative modeling. In the case when the target data distribution has intrinsic structure, such as *group symmetry*, the task of distribution learning can be made more efficient and stable by leveraging the structure of the data. Various empirical studies such as structure-preserving GANs (Birrell et al., 2022), equivariant normalizing flows (Köhler et al., 2020; Garcia Satorras et al., 2021) and equivariant and structure-preserving diffusion models (Hoogeboom et al., 2022; Lu et al., 2024) have shown that symmetry-respecting generative models can effectively learn a group-invariant distribution even with limited data. However, theoretical understanding of these improvements is still limited. To our knowledge, the only work that provides theoretical performance guarantees is Chen et al. (2023c) for group-invariant GANs. In this work, we present new rigorous analysis explaining why score-based generative models (SGMs), or diffusion models (Song & Ermon, 2019; Ho et al., 2020; Song et al., 2020b; Song et al.), can more efficiently learn group-invariant distributions by incorporating the underlying symmetry into the score approximation, as empirically observed in Lu et al. (2024).

**Our contributions.**   We provide the first rigorous error analysis for SGMs with symmetry as well as the first quantitative comparison between data augmentations and incorporating inductive bias of symmetries into generative models. First, by combining recent results relating to the robustness of SGMs with respect to the Wasserstein-1 ($\mathbf{d}_1$) distance (Mimikos-Stamatopoulos et al., 2024) and

the sample complexity of empirical estimations of $\mathbf{d}_1$ for distributions with group symmetry (Chen et al., 2023b; Tahmasebi & Jegelka, 2024), we derive a generalization bound for SGMs with group symmetry to explain the sample efficiency gained when using the symmetry during training. (See Theorem 1 and Theorem 2)

Second, we show that performing standard score-matching, a crucial step in SGM, with respect to any distribution by a $G$-equivariant vector field is equivalent to score-matching with respect to the symmetrized distribution, and that the optimal vector field is exactly the score of the symmetrized distribution (See Theorem 3 and Proposition 1). This provides insights into how to avoid potentially expensive data augmentation by embedding symmetries directly into the score approximation, typically achieved through a $G$-equivariant neural network. Moreover, we compare the impact of non-equivariant score matching via symmetrically augmented datasets with the use of equivariant score matching via the non-augmented datasets using both theory and numerical simulations.

Moreover, we demonstrate the inductive bias of equivariant SGMs using Hamilton-Jacobi-Bellman theory (see Theorem 5).

We adopt a model-form uncertainty quantification (UQ) perspective, attributing errors in equivariant SGMs to the following four sources: $e_1$ – Measurement of the non-equivariance of the learned score function; $e_2$ – Score-matching error with symmetrized vector field; $e_3$ – Sample complexity bound of $\mathbf{d}_1$ with group symmetry; $e_4$ – Error due to early stopping and time horizon.

We show that the generalization error as measured by the expected Wasserstein-1 distance between the generated and target data distributions is bounded by a combination of these four errors above. A particular novelty of our UQ analysis is the quantification of the model-form error $e_1$ of the equivariant structure. This type of UQ perspective was introduced recently for SGMs without structure (Mimikos-Stamatopoulos et al., 2024). Detailed description and discussion of the derived bounds are found in Theorem 2 and Eq. (18).

**Related work.**  Various symmetry-preserving generative models have been proposed such as structure-preserving GANs (Birrell et al., 2022), equivariant normalizing flows (Köhler et al., 2020; Garcia Satorras et al., 2021), equivariant flow matching (Klein et al., 2024), and equivariant diffusion models for molecule generation (Hoogeboom et al., 2022). Theoretical analysis of performance guarantees for such models, to our knowledge, has only been conducted for group-invariant GANs (Chen et al., 2023c). In the context of SGMs, the convergence and generalization of SGMs without group symmetry have been well-studied. The quality of a generated distribution for approximating a target distribution is typically measured by probability divergences and distances. For example, (Chen et al.; Lee et al., 2022; Chen et al., 2023a; Conforti et al., 2023; Oko et al., 2023) prove generalization bounds for TV, $\chi^2$, and $\mathbf{d}_1$ by bounding the KL divergence, which is a stronger divergence. Our results, however, cannot be derived from bounding the KL divergence. The direct $\mathbf{d}_1$ generalization bounds have been derived in (De Bortoli, 2022; Mimikos-Stamatopoulos et al., 2024), but (De Bortoli, 2022) relies on a particular discretization of SGMs. In (Chen et al., 2023b), empirical estimates of the $\mathbf{d}_1$ distance on compact domains of $\mathbb{R}^d$ are shown to obtain a faster convergence assuming the group is finite. Subsequently, (Tahmasebi & Jegelka, 2024) extended the $\mathbf{d}_1$ bound to closed Riemannian manifolds with infinite groups. Our generalization bound for SGM with symmetry is built on the $\mathbf{d}_1$ bounds and UQ perspective for SGMs without structure (Mimikos-Stamatopoulos et al., 2024) and the convergence of the empirical estimations of $\mathbf{d}_1$ distance with group symmetry (Chen et al., 2023b; Tahmasebi & Jegelka, 2024). Recent work (Lu et al., 2024) empirically studies diffusion models with equivariance and proposes various implementations. However, it only provides some guarantees to ensure the generated distribution is $G$-invariant, but no further theory is shown beyond numerical experiments to demonstrate the data efficiency.

The rest of the paper is organized as follows. In Section 2, we review score-based generative models, score-matching objectives, and the notion of group symmetry. We present our theoretical results of generalization bounds in Section 3. Properties of score-matching with equivariant vector fields are presented in Section 4. In Section 5, we discuss the importance of equivariant parametrizations for obtaining a better generalization bound and related insights for practical implementations. We study the inductive bias of equivariant SGMs from the mean-field game perspective in Section 6. In Section 7, we provide numerical experiments that corroborate our theory and insights. We conclude our paper with a discussion in Section 8. All the proofs can be found in the Appendix.

## 2 BACKGROUND

In this section, we introduce group actions and symmetrization operators, and review the score-matching objectives for score-based generative modeling.

### 2.1 GROUP ACTIONS AND SYMMETRIZATION OPERATORS

Let $\Omega$ be the domain, $\mathcal{P}(\Omega)$ the space of probability measures on $\Omega$, and $\mathcal{M}_b(\Omega)$ be the space of bounded measurable functions on $\Omega$. A *group* is a set $G$ equipped with a group product satisfying the axioms of associativity, identity, and invertibility. Given a group $G$ and a set $\Omega$, a map $\theta : G \times \Omega \to \Omega$ is called a *group action on* $\Omega$ if $\theta_g := \theta(g, \cdot) : \Omega \to \Omega$ is an automorphism on $\Omega$ for all $g \in G$, and $\theta_{g_2} \circ \theta_{g_1} = \theta_{g_2 \cdot g_1}, \forall g_1, g_2 \in G$. By convention, we will abbreviate $\theta(g, x)$ as $gx$ throughout the paper.

A function $\gamma : \Omega \to \mathbb{R}$ is called *G-invariant* if $\gamma \circ \theta_g = \gamma, \forall g \in G$. On the other hand, a probability measure $P \in \mathcal{P}(\Omega)$ is called *G-invariant* if $P = (\theta_g)_* P, \forall g \in G$, where $(\theta_g)_* P := P \circ (\theta_g)^{-1}$ is the push-forward measure of $P$ under $\theta_g$. We denote the set of all $G$-invariant distributions on $\Omega$ as $\mathcal{P}_G(\Omega) := \{P \in \mathcal{P}(\Omega) : P \text{ is } G\text{-invariant}\}$.

In this paper, the domain $\Omega$ is bounded; in particular, we focus on the torus $\Omega = R\mathbb{T}^d$ with radius $R$, which is equivalent to a bounded domain with periodic boundary conditions, as considered in (Mimikos-Stamatopoulos et al., 2024). We make the following assumption on $G$.

**Assumption 1.** *G is a group such that the mapping $g : \Omega \to \Omega$ can be written as $g(x) \mapsto A_g x$ for some unitary matrix $A_g \in \mathbb{R}^{d \times d}$ for any $g \in G, x \in \Omega$. That is, any $g \in G$ is a linear isometry.*

Next, we introduce two symmetrization operators from (Birrell et al., 2022), that are useful for our theoretical analysis.

**Symmetrization of functions:** $S_G : \mathcal{M}_b(\Omega) \to \mathcal{M}_b(\Omega)$,

$$S_G[\gamma](x) := \int_G \gamma(gx)\mu_G(\mathrm{d}g) = \mathbb{E}_{\mu_G}[\gamma \circ g(x)], \tag{1}$$

where $\gamma \in \mathcal{M}_b(\Omega)$ and $\mu_G$ is the unique Haar probability measure of $G$.

**Symmetrization of probability measures (dual operator):** $S^G : \mathcal{P}(\Omega) \to \mathcal{P}(\Omega)$, defined for $\gamma \in \mathcal{M}_b(\Omega)$ by

$$\mathbb{E}_{S^G[P]}\gamma := \int_\Omega S_G[\gamma] \, \mathrm{d}P(x) = \mathbb{E}_P S_G[\gamma]. \tag{2}$$

It is shown in (Birrell et al., 2022) that both $S_G$ and $S^G$ define projections. We also abuse the notation that if $P$ evolves with time, then $S^G[P]$ means the symmetrization of $P$ at each time.

We say a vector field $\mathbf{s} : \Omega \times [0, T] \to \mathbb{R}^d$ is *G-equivariant* if

$$\mathbf{s}(gx, t) = A_g \cdot \mathbf{s}(x, t) \tag{3}$$

for any $x \in \Omega$, $g \in G$. It can be easily verified that if $\rho \in \mathcal{P}_G(\Omega)$, then its score $\nabla \log \rho$ is *G-equivariant*. In addition to $S_G$ and $S^G$, we propose

**Symmetrization of vector fields:** $S_G^E : (\Omega \times [0, T] \to \mathbb{R}^d) \to (\Omega \times [0, T] \to \mathbb{R}^d)$,

$$S_G^E[\mathbf{s}](x, t) := \int_G A_g^\top \cdot \mathbf{s}(gx, t)\mu_G(\mathrm{d}g) \tag{4}$$

for any vector field $\mathbf{s}$, which is an extension of formula (12) in (Lu et al., 2024) for finite groups. It can be shown that $S_G^E[\mathbf{s}]$ is *G-equivariant* for any vector field $\mathbf{s}$. The proof can be found in Appendix C. By the definition of equivariance, we immediately have $S_G^E[\mathbf{s}] = \mathbf{s}$ if $\mathbf{s}$ is *G-equivariant*.

The operators $S_G, S^G$, and $S_G^E$ are special types of the Reynolds operator (Rota, 1964).

### 2.2 SCORE-BASED GENERATIVE MODELING

Given a drift term or a vector field $\mathbf{f}(x, t)$, we consider the following forward and backward diffusion processes

$$\mathrm{d}x_s = -\mathbf{f}(x_s, T - s) \, \mathrm{d}s + \sigma(T - s) \, \mathrm{d}W_s, \quad x_0 \sim \pi; \tag{5}$$

$$\mathrm{d}y_t = \Big(\mathbf{f}(y_t, t) + \sigma(t)^2 \nabla \log \eta^\pi(y_t, T-t)\Big)\mathrm{d}t + \sigma(t)\,\mathrm{d}W_t, \quad y_0 \sim m_0,, \tag{6}$$

where $x_s \sim \eta^\pi(\cdot, s)$. Here, $\nabla \log \eta^\pi(x, t)$ is called the score function. It is known from (Anderson, 1982) that if $m_0 = \eta^\pi(\cdot, T)$, then $y_t \sim \eta^\pi(\cdot, T-t)$. In this work, we consider $\mathbf{f} = 0$ and $\sigma(t) = \sqrt{2}$, and the target distribution $\pi \in \mathcal{P}_G(\Omega)$.

Score functions are typically approximated by optimizing parametrized vector fields with respect to the discretization of one of several score-matching objective functions. The *denoising score matching* (DSM) (Vincent, 2011) objective is defined as:

$$\mathcal{J}_D(\eta^\pi, \theta) = \int_0^T \int_\Omega \int_\Omega \Big|\mathbf{s}_\theta - \nabla \log \eta^{x'}\Big|^2 \mathrm{d}\eta^{x'}(s)\,\mathrm{d}\pi(x')\,\mathrm{d}s, \tag{7}$$

where $\eta^{x'}(s)$ denotes the conditional probability from $x'$ at time 0 to $x$ of Eq. (5) at time $s$. In addition, we also introduce two other types of score-matching objectives.

The *explicit score matching* (ESM) objective (Song et al., 2020b), is defined as:

$$\mathcal{J}_E(\rho, \theta) = \int_0^T \int_\Omega |\mathbf{s}_\theta - \nabla \log \rho|^2 \,\mathrm{d}\rho(s)\,\mathrm{d}s, \tag{8}$$

and it is obvious that $\mathcal{J}_E(\rho, \theta) = \mathcal{J}_D(\rho, \theta)$.

The *implicit score matching* (ISM) objective (Song et al., 2020a), is defined as:

$$\mathcal{J}_I(\rho, \theta) := \int_0^T \int_\Omega \Big(|\mathbf{s}_\theta|^2 + 2\nabla \cdot \mathbf{s}_\theta\Big)\mathrm{d}\rho(s)\,\mathrm{d}s, \tag{9}$$

which is more practical for score-matching. By an expansion of the square of the norm, it is easy to verify that $\mathcal{J}_D(\rho, \theta) = \mathcal{J}_E(\rho, \theta) = \mathcal{J}_I(\rho, \theta) + 4\big\|\nabla\sqrt{\rho}\big\|_2^2$ for any $\rho \in \mathcal{P}(\Omega)$. This suggests that the optimal solutions to the DSM, ESM and ISM coincide for the same $\rho$. We also abuse the notation using $\mathcal{J}(\rho, \mathbf{s})$ for a generic vector field $\mathbf{s}$ with an additional subscript on $\mathcal{J}$ when referring to a specific score-matching objective.

## 3 EQUIVARIANT SGMs HAVE IMPROVED $\mathbf{d}_1$ GENERALIZATION BOUNDS

The probability distance we use to measure the generalization error is the Wasserstein-1 distance ($\mathbf{d}_1$), defined as:

$$\mathbf{d}_1(\pi_1, \pi_2) = \sup_{\gamma \in \Gamma}\big\{\mathbb{E}_{\pi_1}[\gamma] - \mathbb{E}_{\pi_2}[\gamma]\big\} \tag{10}$$

for any $\pi_1, \pi_2 \in \mathcal{P}(\Omega)$, where $\Gamma = \mathrm{Lip}_1(\Omega)$ is the set of 1-Lipschitz function on $\Omega$.

In this section, we derive a generalization bound with improved sample complexity in $\mathbf{d}_1$ for learning a $G$-invariant target distribution.

Let $\pi$ be the target data distribution that is $G$-invariant. In SGMs, the generated distribution is $m(T)$, where $m(t)$ follows the denoising diffusion process Eq. (6) with $\nabla \log \rho$ replaced by $\mathbf{s}_\theta$ through score-matching. That is,

$$\partial_t m = \Delta m + 2\,\mathrm{div}(m\mathbf{b}_\theta) \text{ in } \Omega \times (0, T], \quad m(0) = \frac{1}{\mathrm{vol}(R\mathbb{T}^d)} \text{ in } \Omega, \tag{11}$$

where $\mathbf{b}_\theta(x, t) = \mathbf{s}_\theta(x, T-t)$.

In practice, we only have access to finite samples drawn from $\pi$, denoted by $\{z_i\}_{i=1}^N$. Thus, the score-matching or the DSM objective Eq. (7) is often approximated when $\eta^\pi(t)$ is replaced by its kernel density estimate $\eta^N(t)$, where $\eta^N(0) = \pi^N := \frac{1}{N}\sum_{i=1}^N \delta_{z_i}$. Since the kernel estimate does not have a well-defined score at $s = 0$, the DSM objective is often integrated only for $s \in [\epsilon, T]$, an example of early-stopping in SGM (Song et al., 2020b). More specifically, this is equivalent to score-matching for the mollified distribution $\pi^{N,\epsilon} = \pi^N \star \Gamma_\epsilon$, where $\Gamma_\epsilon$ is the heat kernel with time $\epsilon$ and the symbol $\star$ denotes the convolution. In the symmetry-preserving SGM, we consider the symmetrized measure $\pi_G^{N,\epsilon}$, defined as (Tahmasebi & Jegelka, 2024): $\frac{\mathrm{d}\pi_G^{N,\epsilon}}{\mathrm{d}x} = \sum_{l=0}^\infty \exp(-\epsilon\lambda_l)\mu_l\phi_l$, where $\mathrm{d}x$

indicates the uniform measure of $\Omega$, and $(\lambda_l, \phi_l)$ is the pair of the eigenvalues and eigenfunctions of the Laplace-Beltrami operator of $\Omega$, $\mu_l := \frac{1}{N} \sum_{i=1}^{N} \mathbf{1}_G(l)\phi_l(X_i)$, and $\mathbf{1}_G(l) = 1$ if and only if $\phi_l$ is $G$-invariant. In particular, we have $\pi_G^N := \pi^{N,0} = S^G[\pi^N]$. It is evident that $\pi_G^{N,\epsilon} = S^G[\pi^N] \star \Gamma_\epsilon$.

In summary, in the context of SGMs, $\pi^{N,\epsilon} = \pi^N \star \Gamma_\epsilon$ corresponds to early stopping; $\pi_G^N = S^G[\pi^N]$ refers to data augmentations; $\pi_G^{N,\epsilon} = S^G[\pi^N] \star \Gamma_\epsilon$ is the early stopping version of the data-augmented empirical distribution.

Here, we extend the $\mathbf{d}_1$ generalization bound as presented in (Mimikos-Stamatopoulos et al., 2024) to the case when the target distribution is $G$-invariant.

Let $\eta_G^{N,\epsilon} : \Omega \times [0, T] \to [0, \infty)$ be the solution to

$$\begin{cases} \partial_t \rho - \Delta \rho = 0 \text{ in } \Omega \times (0, T], \\ \rho(0) = \pi_G^{N,\epsilon} \text{ in } \Omega, \end{cases} \tag{12}$$

We first prove the finite-sample generalization bound for $\mathbf{d}_1(\pi, m(T))$.

**Theorem 1.** *Assume $\mathcal{J}_D(\eta_G^{N,\epsilon}, \mathbf{s}_\theta) \leq e_{nn}$. Then for $\epsilon < 1$ and up to a dimensional constant $C = C(d) > 0$,*

$$\mathbf{d}_1(\pi, m(T)) \lesssim \sqrt{\epsilon} + R^{3/2}(1 + \sqrt{\|\nabla \mathbf{s}_\theta\|_\infty})\left(Re^{-\frac{wT}{R^2}}\mathbf{d}_1(\pi, \frac{1}{vol(R\mathbb{T}^d)}) + \sqrt{e'_{nn}}\right),$$

*where*

$$e'_{nn} \lesssim e_{nn} + \left(1 - \frac{\log(\epsilon)}{\sqrt{\epsilon}} + \frac{1}{\sqrt{T}} + T\|\mathbf{s}_\theta\|^2_{C^2(\Omega \times [0,T])}\right)\mathbf{d}_1(\pi_G^N, \pi),$$

*and $\pi_G^N$ is the symmetrization of non-symmetric empirical distribution $\pi^N$; i.e., $\pi_G^N = S^G[\pi^N]$.*

**Remark 1.** *The assumption that $\mathcal{J}_D(\eta_G^{N,\epsilon}, \mathbf{s}_\theta) \leq e_{nn}$ implies that the score approximation is trained via DSM with augmented samples. This suggests that equivariant SGMs can be implemented through data augmentations. As we shall see in Sections 5 and 7, a better implementation of equivariant SGMs should rely on equivariant parametrizations of the score function.*

Similar to (Mimikos-Stamatopoulos et al., 2024), we derive the following averaged generalization bound by taking the expectation with respect to the empirical distributions and subsequently applying Jensen's inequality. However, the $G$-invariance of the target distribution $\pi$ provides a significant improvement in the data efficiency in the bounds.

**Theorem 2** (Average bound). *Let $e_{nn}, A > 0$ and assume that for each empirical measure $\pi^N$ consisting of $N$ samples from $\pi$ there exists $\mathbf{s}_\theta$ such that*

$$\mathcal{J}_D(\eta_G^{N,\epsilon}, \mathbf{s}_\theta) \leq e_{nn},$$

*with*

$$\|\mathbf{s}_\theta\|_{C^2(\Omega \times [0,T])} \leq A.$$

*Let $m(T)$ be the generated distribution. Then for sufficiently large $T$, up to a dimensional constant $C$ that only depends on $R$ and $d$ and is independent of random samples or $N$, we have*

$$\mathbb{E}\left[\mathbf{d}_1(\pi, m(T))\right] \lesssim \sqrt{\epsilon} + R^{3/2}(1 + \sqrt{A})\left(Re^{-\frac{wT}{R^2}}\mathbf{d}_1(\pi, \frac{1}{vol(R\mathbb{T}^d)}) + \sqrt{e'_{nn}}\right),$$

*where*

$$e'_{nn} \lesssim e_{nn} + \left(1 - \frac{\log(\epsilon)}{\sqrt{\epsilon}} + \frac{1}{\sqrt{T}} + TA^2\right)\mathbb{E}\left[\mathbf{d}_1(\pi_G^N, \pi)\right].$$

**On the importance of $\mathbf{d}_1$.** The use of $\mathbf{d}_1$ distance on both sides of our generalization bounds has two key implications:

(1) We can take advantage of the $G$-invariance of $\pi$ and improve data efficiency since $\mathbf{d}_1$ allows gains on $\mathbb{E}[\mathbf{d}_1(\pi_G^N, \pi)]$. First, it is shown in (Chen et al., 2023b) that on bounded domains of $\mathbb{R}^d$, we have

$$
\mathbb{E}[\mathbf{d}_1(\pi_G^N, \pi)] \lesssim
\begin{cases}
\left(\frac{1}{|G|N}\right)^{1/d} & \text{if } d \geq 3, \\
\left(\frac{1}{|G|N}\right)^{1/2} \log N & \text{if } d = 2, \\
\frac{\text{diam}(\Omega/G)}{N^{1/2}} & \text{if } d = 1,
\end{cases}
\tag{13}
$$

if $G$ is finite. Later, (Tahmasebi & Jegelka, 2024) extend it to closed Riemannian manifolds with possibly infinite $G$ such that $\mathbb{E}[\mathbf{d}_1(\pi_G^N, \pi)] \lesssim \left(\frac{\text{vol}(\Omega/G)}{N}\right)^{1/d^*}$, where $\text{vol}(\Omega/G)$ is the volume of the quotient space $\Omega/G$ and $d^* = \dim(\Omega/G) \geq 3$. This sample complexity gain cannot be derived for the KL or other $f$-divergences without additional regularization.

(2) The $\mathbf{d}_1$ bounds in Theorem 1 and Theorem 2 remain well-defined and meaningful even when the target probability distribution does not have a density. In particular, Theorem 2 has the following corollary when the target distribution is supported on a smooth submanifold $\mathcal{M} \subset \Omega$.

**Corollary 1.** *Follow the same assumption and quantities as in Theorem 2, and assume that $\pi$ is supported on a closed submanifold $\mathcal{M} \subset \Omega$, and $G$ admits a unitary representation in $\Omega$ as in Assumption 1. Then up to a dimensional constant $C > 0$ that also depends on $\mathcal{M}$, such that*

$$
\mathbb{E}\left[\mathbf{d}_1(\pi, m(T))\right] \lesssim \sqrt{\epsilon} + R^{3/2}(1+\sqrt{A})\left(Re^{-\frac{wT}{R^2}}\mathbf{d}_1(\pi, \frac{1}{vol(R\mathbb{T}^d)}) + \sqrt{e'_{nn}}\right),
$$

*where*

$$
e'_{nn} \lesssim e_{nn} + \left(1 - \frac{\log(\epsilon)}{\sqrt{\epsilon}} + \frac{1}{\sqrt{T}} + TA^2\right)\left(\frac{vol(\mathcal{M}/G)}{N}\right)^{1/d^*},
$$

*where $vol(\mathcal{M}/G)$ is the volume of the quotient space $\mathcal{M}/G$ and $d^* = dim(\mathcal{M}/G) \geq 3$, and $\mathbf{d}_1$ here denotes the Wasserstein-1 distance on $\Omega$.*

Corollary 1 illustrates that the convergence rate in terms of the number of samples $N$ in the generalization bound can be improved from $d^{-1}$ to $d^{*-1}$ in the exponent, which depends on the dimension of the quotient space $\mathcal{M}/G$.

# 4 EQUIVARIANT PARAMETRIZATIONS RESTORE INTRINSIC EQUIVARIANCE OF SGMS

Theorem 1 and Theorem 2 do not explicitly convey the significance of equivariant vector fields in score matching. First, we illustrate the importance of equivariance from a Hamilton-Jacobi-Bellman (HJB) perspective in Section 6 by showing that SGMs are *intrinsically* equivariant. Second, we highlight the role of $G$-equivariant vector fields (typically parameterized by neural networks) in score matching, an aspect that has only been addressed experimentally in previous studies. Our rigorous results indicate that it is sufficient to perform score matching with $G$-equivariant vector fields in relation to an unsymmetrized distribution. This approach will be particularly beneficial when we only have a finite set of *unaugmented* samples (i.e., a non-symmetric empirical distribution drawn from an invariant distribution). This latter aspect will be discussed in detail in Section 4.1, Section 5 and tested in Section 7.

## 4.1 PROPERTIES OF SCORE-MATCHING WITH EQUIVARIANT VECTOR FIELDS

First, we show that for any distribution $\rho$, the ISM objective when restricted to $G$-equivariant vector fields, is equivalent to the ISM objective with respect to its symmetrized counterpart. Second, we prove that using equivariant vector fields can reduce the DSM error for $G$-invariant distributions.

**Theorem 3.** *Consider the ISM problem in Eq. (9), in which $\rho$ is not necessarily $G$-invariant. Then for any $G$-equivariant vector field $\mathbf{s}$, we have*

$$
\mathcal{J}_I(\rho, \mathbf{s}) = \mathcal{J}_I(S^G[\rho], \mathbf{s}).
$$

**Remark 2.** *Theorem 3 is important for practical implementations, in the sense that the optimal equivariant vector field can be obtained by score-matching for raw data **without** data augmentation. We will demonstrate this point in our numerical simulations in Section 7.*

Moreover, for the ESM (or equivalently, the DSM) problem of a generic probability measure, the $G$-equivariant minimizer is exactly the score of the symmetrized probability measure, namely:

**Proposition 1.** *Consider the ESM problem in Eq. (8), in which $\rho$ is not necessarily $G$-invariant. Denote by $V_G \subset \Omega \times [0,T] \to \mathbb{R}^d$, the subspace of $G$-equivariant vector fields. Then we have*

$$\arg\min_{\mathbf{s} \in V_G} \mathcal{J}_E(\rho, \mathbf{s}) = \nabla_x \left[ \log \left( S^G[\rho] \right) \right].$$

We propose the following definition as an error quantification for the non-equivariance of a vector field with respect to a $G$-invariant measure $\rho \in \mathcal{P}_G(\Omega) \times [0,T]$.

**Definition 1** (Deviation from equivariance). *The deviation from equivariance (DFE) of a vector field $\mathbf{s}$ with respect to $\rho \in \mathcal{P}_G(\Omega) \times [0,T]$ is defined as*

$$DFE(\rho, \mathbf{s}) := \int_0^T \int_\Omega \left| \mathbf{s} - S_G^E[\mathbf{s}] \right|^2 \mathrm{d}\rho(s)\,\mathrm{d}s. \tag{14}$$

It is evident that $DFE(\rho, \mathbf{s}) = 0$ if $\mathbf{s}$ is $G$-equivariant. Given this definition, we obtain the following decomposition of the ESM and DSM objectives.

**Theorem 4.** *For any $\rho \in \mathcal{P}_G(\Omega) \times [0,T]$ and any vector field $\mathbf{s}$, we have*

$$\mathcal{J}_E(\rho, \mathbf{s}) = DFE(\rho, \mathbf{s}) + \mathcal{J}_E(\rho, S_G^E[\mathbf{s}]). \tag{15}$$

*As DSM and ESM are equivalent objectives, we readily have*

$$\mathcal{J}_D(\rho, \mathbf{s}) = DFE(\rho, \mathbf{s}) + \mathcal{J}_D(\rho, S_G^E[\mathbf{s}])\,, \text{ for any } \rho \in \mathcal{P}_G(\Omega) \times [0,T]\,. \tag{16}$$

Finally, the following proposition indicates that for any learned distribution $\eta$, its symmetrized counterpart $S^G[\eta]$ is always closer to the $G$-invariant target distribution $\pi$ in the $\mathbf{d}_1$ sense. The $G$-invariance of the generated distribution is guaranteed by the $G$-equivariant vector field $\mathbf{s}_\theta$ (see Corollary 2).

**Proposition 2.** *For any $\eta, \pi \in \mathcal{P}(\Omega)$, and $\pi$ is $G$-invariant, we have*

$$\mathbf{d}_1(\eta, \pi) \geq \mathbf{d}_1(S^G[\eta], \pi).$$

## 5 THE SIGNIFICANCE OF EQUIVARIANT VECTOR FIELDS IN SGMs

With the theoretical results established in Section 3 and Section 4, we can now focus on providing quantitative comparisons between equivariant vector fields and data augmentations. Our strategy relies on making the generalization bound in Theorem 2 as small as possible. In particular, we take a closer look at the terms $e_{nn}$ and $\mathbb{E}[\mathbf{d}_1(\pi_G^N, \pi)]$, which can be improved by selecting an appropriate structure for the vector field or by implementing data augmentations.

The assumption $\mathcal{J}_D(\eta_G^{N,\epsilon}, \mathbf{s}_\theta) \leq e_{nn}$ in Theorem 2 refers to the error of DSM with augmented data. Technically, this assumption ensures the same generalization bounds derived in Theorem 1 and Theorem 2, regardless of whether the vector field $\mathbf{s}_\theta$ is $G$-equivariant or not. Note also that the gain in $\mathbb{E}[\mathbf{d}_1(\pi_G^N, \pi)]$ (see the paragraph after Theorem 2 for the sample complexity gain) is not affected no matter whether we use equivariant vector fields. However, $\mathcal{J}_D(\eta_G^{N,\epsilon}, \mathbf{s}_\theta)$ or $e_{nn}$ does depend on the structure of vector fields and can be improved accordingly as we see next.

- **Data augmentation without equivariant structure:** If we perform data augmentations without using equivariant vector fields, then we have to pay the cost of data augmentations. Moreover, by Theorem 4,

$$e_{nn} = \mathcal{J}_D(\eta_G^{N,\epsilon}, \theta) = \mathrm{DFE}(\eta_G^{N,\epsilon}, \mathbf{s}_\theta) + \mathcal{J}_D(\eta_G^{N,\epsilon}, S_G^E[\mathbf{s}_\theta])\,, \tag{17}$$

therefore $e_{nn}$ has a lower bound of $\mathrm{DFE}(\eta_G^{N,\epsilon}, \mathbf{s}_\theta)$ that measures the distortion of vector fields from equivariance, which can be large if the vector fields are highly "non-equivariant".

- **Equivariant structure without data augmentation:** On the contrary, if we simply use equivariant vector fields without data augmentations, by Theorem 3, we can automatically obtain the score approximations of $\eta_G^{N,\epsilon}$ by simply solving the ISM objective of *unaugmented* samples $\eta^{N,\epsilon}$. Thus, the assumption $\mathcal{J}_D(\eta_G^{N,\epsilon}, \theta) \leq e_{nn}$ is valid in practice. The main difference with the simple data augmentation case discussed above is that here, due to restricting the SGM on equivariant vector fields, we have $\text{DFE}(\eta_G^{N,\epsilon}, \mathbf{s}_\theta) = 0$. Therefore, the term $e_{nn}$ in the generalization bounds can be made as small as possible, assuming the equivariant NN can be parametrized efficiently and has sufficient expressive power, which has been verified empirically in, e.g., Cohen & Welling (2016); Lu et al. (2024).

To summarize, the generalization bound in Theorem 2 can be re-written as

$$\mathbb{E}\left[\mathbf{d}_1(\pi, m(T))\right] \lesssim \text{DFE}(\eta_G^{N,\epsilon}, \mathbf{s}_\theta) + \mathcal{J}_D(\eta_G^{N,\epsilon}, S_G^E[\mathbf{s}_\theta]) + \mathbb{E}[\mathbf{d}_1(\pi_G^N, \pi)] + C(\epsilon, T), \quad (18)$$

where $C(\epsilon, T)$ accounts for the error from early stopping and time horizon, and is independent of the equivariance structure or data augmentations we are studying. This suggests that while data augmentations can provide gains in $\mathbb{E}[\mathbf{d}_1(\pi_G^N, \pi)]$, in order to further minimize the generalization error, one should make $\text{DFE}(\eta_G^{N,\epsilon}, \mathbf{s}_\theta) = 0$; that is, applying $G$-equivariant vector fields.

To be more specific, when the group is finite, we can always augment the data, and we can also design equivariant NNs, at least using the symmetrization operator $S_G^E$. Based on our theory, equivariant models produce smaller generalization errors as they have precisely zero DFE. For infinite groups, we can not perform a complete and exact data augmentation. However, it is possible to design equivariant NNs for continuous groups, though the problem is still open to our knowledge. Moreover, once we have such architectures, we can obtain data augmentation for free by Theorem 3.

# 6 HJB DESCRIBES THE INDUCTIVE BIAS OF EQUIVARIANT SGMs

We use the connections between SGMs and PDE theory to provably show that score-based generative models are intrinsically equivariant under relatively mild assumptions. Score-based generative models have been shown to be well-posed through their connections with stochastic optimal control and mean-field games (MFGs) (Berner et al., 2022; Zhang & Katsoulakis, 2023; Zhang et al., 2024). In Zhang & Katsoulakis (2023); Zhang et al. (2024), it was shown that score-based generative models are solutions of a mean-field game, more specifically, one that corresponds with the Wasserstein proximal of the cross-entropy. The peculiar structure of cross-entropy is why SGMs can be trained by score-matching alone. The MFG is an infinite-dimensional optimization problem

$$\min_{v, \rho} \left\{ -\int_\Omega \log \pi(x) \rho(x, T) dx + \int_0^T \int_\Omega \left[ \frac{1}{2} \|v\|^2 - \nabla \cdot f \right] \rho(x, t) dx dt \right\} \quad (19)$$

$$\text{s.t. } \partial_t \rho + \nabla \cdot ((f + \sigma v)\rho) = \frac{\sigma^2}{2} \Delta \rho, \; \rho(x, 0) = \eta(x, T).$$

The density of particles evolve according to the controlled Fokker-Planck equation. The terminal cost is equivalent to the cross entropy of $\pi$ with respect to the terminal density $\rho(x, T)$. The running cost is, via the Benamou-Brenier formulation of optimal transport, the Wasserstein-2 distance with a state cost $-\nabla \cdot f$.

The solution of the MFG optimization problem is characterized by its optimality conditions, which are a pair of nonlinear partial differential equations.

$$\begin{cases} -\partial_t U - f^\top \nabla U + \frac{1}{2} |\sigma \nabla U|^2 + \nabla \cdot f = \frac{\sigma^2}{2} \Delta U \\ \partial_t \rho + \nabla \cdot (\rho(f - \sigma^2 \nabla U)) = \frac{\sigma^2}{2} \Delta \rho \\ U(x, T) = -\log \pi(x), \; \rho(x, 0) = e^{-U(x,0)}. \end{cases} \quad (20)$$

This first equation is a Hamilton-Jacobi-Bellman equation, which determines the optimal velocity field $v^*(x, t) = -\sigma \nabla U$ for the second equation, a controlled Fokker-Planck. By a Hopf-Cole

(logarithmic) transformation, this pair of PDEs is equivalent to the noising-denoising SDE system. Let $U(x,t) = -\log \eta(x, T-t)$, then for $s = T - t$, we have

$$
\begin{cases}
\dfrac{\partial \eta}{\partial s} = -\nabla \cdot (f\eta) + \dfrac{\sigma^2}{2}\Delta \eta \\[2mm]
\dfrac{\partial \rho}{\partial t} = -\nabla \cdot (\rho(f + \sigma^2 \nabla \log \eta(x, T-t))) + \dfrac{\sigma^2}{2}\Delta \rho \\[2mm]
\eta(x, 0) = \pi(x), \quad \rho(x, 0) = \eta(x, T).
\end{cases}
$$

We can then see that the optimal velocity field has the form $v^\star(x,t) = -\sigma(t)\nabla U(x,t) = \sigma(T-t)\nabla \log \eta(x, T-t)$, which is precisely related linearly with respect to the score function of the forward noising process.

**Theorem 5.** *Consider the score-based generative model given by the equivalent MFG Eq. (19) and let $U$ be the solution to the HJB equation in Eq. (20). Assume the target data distribution $\pi$ is $G$-invariant and that the drift in the noising dynamics is $G$-equivariant. Then we have that the corresponding score function is $G$-equivariant, namely*

$$
\mathbf{s}^*(x,t) = -\nabla U(x,t) = \underset{\mathbf{s} \in \Omega \times [0,T] \to \mathbb{R}^d}{\arg\min}\ \mathcal{J}_E(\rho, \mathbf{s}) \in V_G\,, \tag{21}
$$

*where we denote by $V_G \subset \Omega \times [0,T] \to \mathbb{R}^d$, the subspace of $G$-equivariant vector fields.*

The MFG perspective is useful as the proof for this theorem immediately follows from basic uniqueness results from PDE theory. This theorem states that, mathematically, SGMs are symmetry-preserving for invariant target measures when the drift function also preserves the same symmetry. This result holds for *any* group $G$. The most trivial case is when $f = 0$.

**Remark 3** (Equivariant inductive bias). *In the SGM algorithm the optimal vector field $\mathbf{s}^*(x,t)$ is the score and is learned as part of the algorithm. Therefore, this theorem shows that the corresponding neural network for the approximation of $\mathbf{s}^*(x,t)$ should be parameterized in a way that is also $G$-equivariant, thus incorporating in the algorithm the inherent equivariant (structural) inductive bias of Theorem 5.*

## 7 NUMERICAL EXAMPLE

We provide a simple numerical experiment to validate the basic results of our theory. The primary purpose is to emphasize minimizing the score-matching objective with respect to a non-symmetric sample of $G$-invariant distribution $\pi$ within a class of $G$-equivariant vector fields is better than just augmenting the data through group actions, as is indicated by our analysis encapsulated in the generalization bound Eq. (18).

We consider a mixture of 4 Gaussians centered at $[\pm 5, \pm 5]$ in $\mathbb{R}^2$. The group is generated by the action of moving from one center to the next. We report the $\mathbf{d}_1$ distance between the generated distribution and the target distribution. We consider four experimental setups: the first case (**Equivariant, not augmented**) is where the score network is parametrized to be $G$-equivariant by parametrizing it as

$$
\mathbf{s}_\theta^G(x,t) = \frac{1}{|G|}\sum_{g \in G} A_g^\top \mathbf{s}_\theta(A_g x, t), \tag{22}
$$

where $|G| = 4$ is the order of the group. The score is trained on $N_t$ samples that are not augmented. The second case (**Equivariant, augmented**) is where the score network is parametrized as in Eq. (22), and is trained on data that is augmented by applying each group action on each training sample (hence effectively $4 \times N_t$ samples). The third case (**Non-equivariant, augmented**) is where the network $\mathbf{s}_\theta$ is trained directly but on augmented training data. The fourth case (**Non-equivariant, not augmented**) is where the network $\mathbf{s}_\theta$ is trained directly and the training data is not augmented. For each case, the function $\mathbf{s}_\theta$ is parametrized via a fully-connected neural network with 3 hidden layers and 32 nodes per layer. It is trained over 10000 iterations via stochastic gradient descent, where the batch size is $N_{batch} = 32$. For $N_t = 10$, we sample with replacement in the SGD. We compute the Wasserstein-1 distance using its dual form $\mathbf{d}_1(\eta, \pi) = \sup\left\{\mathbb{E}_\eta[\psi] - \mathbb{E}_\pi[\psi] : \psi \in \mathrm{Lip}_1(\Omega)\right\}$. The function $\psi$ is parametrized by a fully-connected neural network with two hidden layers with 64 nodes per layer. Spectral normalization (Miyato et al., 2018) is applied to enforce the Lipschitzness of $\psi$.

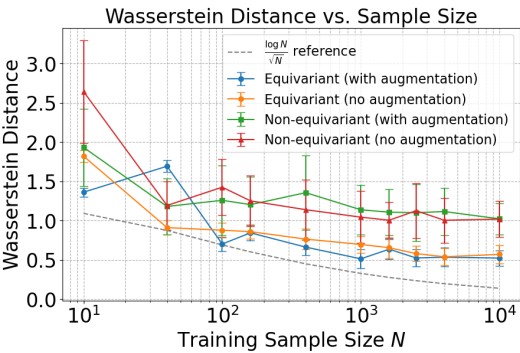

Figure 1: Wasserstein distance as a function of training sample size.

For each value of $N_t$ we perform 25 runs of each method. The mean and standard deviation of the 25 runs are reported in Table 1 and in Figure 1. Notice that the equivariant case consistently performs better than the data-augmented case, which corroborates our theoretical analysis. Moreover, the results suggest that training a non-equivariant score network on augmented data may not necessarily produce a superior model to the case when the data is not augmented.

In Figure 2, we show the generated samples of each case when $N_t = 40$. Observe that the only way to consistently produce an invariant generated distribution is to have use an equivariant score approximation. Moreover, note that the reduction of $\mathbf{d}_1$ becomes marginal for large $N_t$ as other errors in the Theorem 2 are independent of the number of training samples.

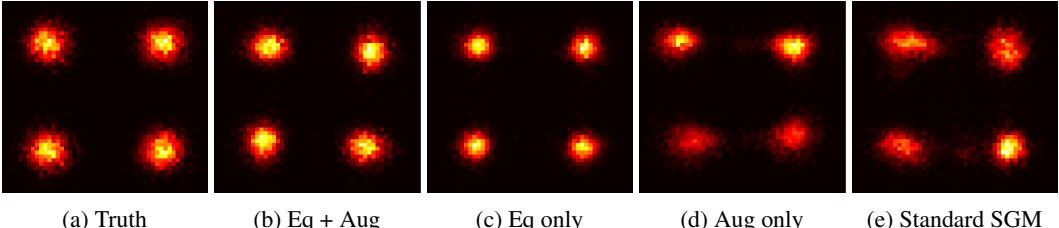

| (a) Truth | (b) Eq + Aug | (c) Eq only | (d) Aug only | (e) Standard SGM |

Figure 2: Score-based generative modeling for a simple 2D mixture of Gaussians. Training dataset is of size $N_t = 40$.

Table 1: $\mathbf{d}_1$ value for a 2d Gaussian mixture

| $N_t$ | Equivariant, augmented | Equivariant, not augmented | Non-equivariant, augmented | Non-equivariant, not augmented |
|---|---|---|---|---|
| 10 | $1.36 \pm 0.06$ | $1.82 \pm 0.08$ | $1.93 \pm 0.49$ | $2.64 \pm 0.65$ |
| 100 | $0.70 \pm 0.09$ | $0.88 \pm 0.10$ | $1.26 \pm 0.45$ | $1.43 \pm 0.35$ |
| 1000 | $0.51 \pm 0.12$ | $0.70 \pm 0.11$ | $1.14 \pm 0.32$ | $1.04 \pm 0.33$ |
| 10000 | $0.52 \pm 0.10$ | $0.57 \pm 0.12$ | $1.02 \pm 0.20$ | $1.02 \pm 0.23$ |

## 8 CONCLUSION AND FUTURE WORK

We rigorously show that SGMs can learn distributions with symmetries efficiently with equivariant score approximations. Compared to data augmentations, using equivariant vector fields for score-matching has the additional gain of reducing the score approximation error without the need to augment the dataset. Numerical experiments further verify this theoretical result. Certain directions are still unexplored in the present work. For instance, it would be valuable to explore the architecture of equivariant neural networks to ensure they possess sufficient expressive power while maintaining a manageable number of parameters with reduced training cost, as in the group equivariant convolutional neural networks proposed in (Cohen & Welling, 2016) for discrete groups or even continuous groups, which remains an open problem. Furthermore, our analysis does not account for the time discretization of SGMs, and it could be interesting to incorporate this aspect or explore symmetry-preserving numerical integrators within the theoretical framework. Another extension of our work would be to consider the domain as $\mathbb{R}^d$, with the forward process being, for instance, an Ornstein–Uhlenbeck process or other nonlinear processes (Birrell et al., 2024; Singhal et al., 2024).

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

## A  PROOF OF THEOREM 1

We also define the $G$-regularized Wasserstein-1 distance ($\mathbf{d}_1^G$) as:

$$\mathbf{d}_1^G(\pi_1, \pi_2) = \sup_{\gamma \in \Gamma_G^{inv}} \left\{ \mathbb{E}_{\pi_1}[\gamma] - \mathbb{E}_{\pi_2}[\gamma] \right\}, \tag{23}$$

where $\Gamma_G^{inv}$ is the subset of $\Gamma$ that consists of all $G$-invariant 1-Lipschitz functions.

The following theorem is adapted from Theorem 3.1 in (Mimikos-Stamatopoulos et al., 2024). Here we prove a version with group symmetry. The main difference is that the test function is now restricted to the class of $G$-invariant 1-Lipschitz functions, which is guaranteed by the equivariance of $b^1$.

**Theorem 6** (Wasserstein Uncertainty Propagation). *Let $\Omega = R\mathbb{T}^d$. Let $G$-equivariant vector fields $b^1, b^2 : \Omega \times [0, T] \to \mathbb{R}^d$ be given with $\left\|\nabla b^1\right\|_\infty < \infty$ and $m_1, m_2 \in \mathcal{P}_G(\Omega)$. If $m^i$ for $i = 1, 2$ are given by*

$$\partial_t m^i - \Delta m^i - \operatorname{div}(m^i b^i) = 0, \ m^i(0) = m_i. \tag{24}$$

*Then up to a universal constant $C > 0$, we have*

$$\mathbf{d}_1^G(m^2(T), m^1(T)) = \mathbf{d}_1(m^2(T), m^1(T)) \le CR^{\frac{3}{2}}(1 + \sqrt{\|\nabla b^1\|_\infty})(\mathbf{d}_1^G(m_2, m_1) + \epsilon_1),$$

*if*

$$\left\|b^2 - b^1\right\|_{L^2(m^2)} := \left(\int_0^T \int_\Omega \left|(b^2 - b^1)(x, t)\right|^2 m^2(t, x) \, \mathrm{d}x \, \mathrm{d}t\right)^{\frac{1}{2}} \le \epsilon_1.$$

*Proof.* The measure $\lambda = m^1 - m^2$ satisfies the PDE

$$\partial_t \lambda - \Delta \lambda - \operatorname{div}(\lambda b^1 + m^2(b^1 - b^2)) = 0 \text{ in } \Omega \times (0, T), \quad \lambda(0) = m_2 - m_1 \text{ in } \Omega. \tag{25}$$

Let $\phi : \Omega \times [0, T] \to \mathbb{R}$ be a test function in space and time. We integrate in space and time against the PDE Eq. (25) and apply integration by parts to obtain

$$\int_\Omega \lambda(x, T)\phi(x, T) - \lambda(x, 0)\phi(x, 0) \, \mathrm{d}x + \int_0^T \int_\Omega \lambda(-\partial_t \phi - \Delta \phi + b^1 \cdot \nabla \phi) \, \mathrm{d}x \, \mathrm{d}t \tag{26}$$

$$+ \int_0^T \int_\Omega m^2 \nabla \phi \cdot (b^1 - b^2) \, \mathrm{d}x \, \mathrm{d}t = 0$$

Notice that if we choose the test function $\phi$ to satisfy the Kolmogorov backward equation (KBE)

$$-\partial_t \phi - \Delta \phi + b^1 \cdot \nabla \phi = 0 \text{ in } \Omega \times [0, T], \quad \phi(x, T) = \psi(x) \text{ in } \Omega \tag{27}$$

with terminal condition $\psi \in \mathcal{F}$, then from Eq. (26), we have

$$\int_\Omega \lambda(x, T)\psi(x) \, \mathrm{d}x = \int_\Omega \lambda(x, 0)\phi(x, 0) \, \mathrm{d}x + \int_0^T \int_\Omega m^2(t)\nabla \phi(x, t) \cdot (b^2 - b^1)(t) \, \mathrm{d}x \, \mathrm{d}t. \tag{28}$$

Let $\mathcal{F}$ be the set of $G$-invariant 1-Lipschitz functions on $\Omega$. Taking the supremum over $\mathcal{F}$ we have

$$\mathbf{d}_1^G(m^2(T), m^1(T)) \le \sup_{\psi \in \mathcal{F}}\left|\int_\Omega \lambda(x, 0)\phi(x, 0) \, \mathrm{d}x\right| + \sup_{\psi \in \mathcal{F}}\left|\int_0^T \int_\Omega m^2 \nabla \phi \cdot (b^2 - b^1) \, \mathrm{d}x \, \mathrm{d}t\right|. \tag{29}$$

Also recall that $\phi$ is related to $\psi$ via the KBE Eq. (27). We first show that $\phi(x, t)$ is always $G$-invariant for any $t \in [0, T]$ as long as $\psi$ is $G$-invariant. Indeed, if we perform a Hopf-Cole transform $u = -2 \log \phi$, then Eq. (27) is equivalent to the Hamilton-Jacobi-Bellman (HJB) equation for $u$

$$-\partial_t u - \Delta u + \frac{1}{2}|\nabla u|^2 + V \cdot \nabla \phi = 0, \ u(x, T) = -2\log(\psi(x)). \tag{30}$$

On the other hand, it can easily be verified that $h(x, t) = u(gx, t)$ also satisfies Eq. (30) for any $g \in G$ since $A_g$ is unitary and $b^1$ is $G$-equivariant. The existence and uniqueness of the solution to Eq. (30) (Evans, 2022) guarantees that $h(x, t) = u(x, t)$ is $G$-invariant for any $t \in [0, T)$ and therefore we have $\phi(x, t) = \phi(gx, t)$ for any $g \in G$ and $t \in [0, T)$. The rest of the proof, i.e., the gradient estimate of $\phi$ is exactly the same as that of Theorem 3.1 in (Mimikos-Stamatopoulos et al., 2024) since any $\psi \in \mathcal{F}$ is 1-Lipschitz. $\square$

**Corollary 2.** *Suppose a probability measure $m(x, t)$ evolves according to the KBE Eq. (27). That is,*

$$-\partial_t m - \Delta m + V \cdot \nabla m = 0 \text{ in } \Omega \times [0, T], \quad m(x, T) = m_0 \text{ in } \Omega \tag{31}$$

*where the vector field $V$ is $G$-equivariant and the terminal measure $m_0$ is $G$-invariant. Then $m(x, t)$ is $G$-invariant for all $t \in [0, T)$.*

*Proof.* By a change of variable $t \mapsto -t$ in the KBE Eq. (27), the statement follows the proof after Eq. (30). $\qquad\square$

The following proposition shows that for empirical measures, the action of diffusion and symmetrization are commutable.

**Proposition 3.** $S^G[\pi^{N,\epsilon}] = S^G[\pi^N] \star \Gamma_\epsilon$.

*Proof.* For any $\gamma \in \mathcal{M}_b(\Omega)$, we have

$$\mathbb{E}_{S^G[\pi^{N,\epsilon}]}\gamma = \mathbb{E}_{\pi^{N,\epsilon}}S_G[\gamma]$$

$$= \int_\Omega \pi^N \star \Gamma_\epsilon S_G[\gamma]\,\mathrm{d}x$$

$$= \int_\Omega \int_G \int_\Omega \pi^N(y)\Gamma_\epsilon(x-y)\,\mathrm{d}y\gamma(gx)\mu_G(\mathrm{d}g)\,\mathrm{d}x$$

$$= \int_\Omega \int_G \int_\Omega \pi^N(y)\Gamma_\epsilon(g^{-1}x-y)\,\mathrm{d}y\gamma(x)\mu_G(\mathrm{d}g)\,\mathrm{d}x \quad \text{(since the Jacobian of } g \text{ is unitary)}$$

$$= \int_\Omega \int_G \int_\Omega \pi^N(g^{-1}y)\Gamma_\epsilon(g^{-1}x-g^{-1}y)\,\mathrm{d}y\gamma(x)\mu_G(\mathrm{d}g)\,\mathrm{d}x$$

$$= \int_\Omega \int_G \int_\Omega \pi^N(g^{-1}y)\Gamma_\epsilon(x-y)\,\mathrm{d}y\gamma(x)\mu_G(\mathrm{d}g)\,\mathrm{d}x \quad \text{(due to the property of the heat kernel)}$$

$$= \int_\Omega \int_\Omega \int_G \pi^N(g^{-1}y)\mu_G(\mathrm{d}g)\Gamma_\epsilon(x-y)\,\mathrm{d}y\gamma(x)\,\mathrm{d}x$$

$$= \mathbb{E}_{S^G[\pi^N]\star\Gamma_\epsilon}\gamma.$$

$\qquad\square$

We decompose $\mathbf{d}_1(\pi, m(T))$ as follows

$$\mathbf{d}_1(\pi, m(T)) \leq \mathbf{d}_1(\pi, \pi^\epsilon) + \mathbf{d}_1(\pi^\epsilon, m(T)). \tag{32}$$

For the early stopping error, by the proof of Theorem 3.3 in (Mimikos-Stamatopoulos et al., 2024), we have $\mathbf{d}_1(\pi, \pi^\epsilon) \leq C\sqrt{\epsilon}$, where $C$ only depends on the dimension $d$. To bound the second term in Eq. (32), we define $\eta^{\pi,\epsilon} : [0,T] \times R\mathbb{T}^d \to \mathbb{R}$ given by

$$\begin{cases} \partial_t \eta^{\pi,\epsilon} - \Delta\eta^{\pi,\epsilon} = 0 \text{ in } R\mathbb{T}^d \times (0,T), \\ \eta^{\pi,\epsilon}(0) = \pi^\epsilon \text{ in } R\mathbb{T}^d. \end{cases} \tag{33}$$

Moreover, we define the drift

$$\mathbf{b}^{\pi,\epsilon}(x,t) := \nabla\log(\eta^{\pi,\epsilon})(x, T-t)$$

and let $m^\epsilon(x,t) = \eta^{\pi,\epsilon}(x, T-t)$ which satisfies

$$\begin{cases} \partial_t m^\epsilon = \Delta m^\epsilon + 2\operatorname{div}(m^\epsilon \mathbf{b}^{\pi,\epsilon}), \\ m^\epsilon(0) = \eta^{\pi,\epsilon}(T). \end{cases} \tag{34}$$

Then by applying Theorem 6, we have

$$\mathbf{d}_1(\pi^\epsilon, m(T)) = \mathbf{d}_1(m^\epsilon(T), m(T))$$

$$\lesssim R^{\frac{3}{2}}(1 + \sqrt{\|\mathbf{b}_\theta\|_\infty})\left(\mathbf{d}_1(m^\epsilon(0), \frac{1}{\operatorname{vol}(R\mathbb{T}^d)}) + \|\mathbf{b}^{\pi,\epsilon} - \mathbf{b}_\theta\|_{L^2(m^\epsilon)}\right),$$

where we use the symbol '$\lesssim$' to absorb the universal universal constant $C$ defined in Theorem 6.

By proposition A.3 in (Mimikos-Stamatopoulos et al., 2024), we have

$$\mathbf{d}_1(m^\epsilon(0), \frac{1}{\operatorname{vol}(R\mathbb{T}^d)}) = \mathbf{d}_1(\eta^{\pi,\epsilon}(T), \frac{1}{\operatorname{vol}(R\mathbb{T}^d)}) \leq CRe^{-\frac{wT}{R^2}}\mathbf{d}_1(\pi^\epsilon, \frac{1}{\operatorname{vol}(R\mathbb{T}^d)}).$$

It remains to show the following bound

$$\|\mathbf{b}^{\pi,\epsilon} - \mathbf{b}_\theta\|^2_{L^2(m^\epsilon)} = \mathcal{J}_D(\eta^{\pi,\epsilon}, \theta) \le e'_{nn} = e_{nn} + C\left(1 - \frac{\log\epsilon}{\sqrt{\epsilon}} + \frac{1}{\sqrt{T}} + T\|\mathbf{s}_\theta\|^2_{C^2(\Omega\times[0,T])}\right)\mathbf{d}_1(\pi_G^N, \pi). \tag{35}$$

In the rest part of this section, we prove Eq. (35). The proof is based on the structure of Section 8 in (Mimikos-Stamatopoulos et al., 2024).

We denote by $\rho^{m_0} : \Omega \times [0,T] \to [0,\infty)$ the solution to

$$\begin{cases} \partial_t \rho^{m_0} - \Delta\rho^{m_0} = 0 \text{ in } \Omega \times (0,T], \\ \rho^{m_0}(0) = m_0 \text{ in } \Omega. \end{cases} \tag{36}$$

**Lemma 1** (Proposition 8.1 in (Mimikos-Stamatopoulos et al., 2024)). *Let $m_0$ be a probability density in $\Omega$, such that $m_0\log(m_0) \in L^1(\Omega)$ and $\rho : \Omega \times [0,T] \to \mathbb{R}$ be given by Eq. (36). Then we have*

$$4\|\nabla\sqrt{\rho}\|^2_2 = \int_\Omega m_0\log(m_0) - \rho(T)\log(\rho(T))\,\mathrm{d}x.$$

**Lemma 2** (Proposition 8.2 in (Mimikos-Stamatopoulos et al., 2024)). *Let $\pi^i$ ($i = 1,2$) denote two probability measures in $\Omega$ such that $\|\pi^i\log(\pi^i)\|_1 < \infty$ and $\rho^i$ the corresponding solutions to Eq. (36). Then there exists a dimensional constant $C > 0$ such that*

$$\left|\mathcal{J}_I(\rho^2, \theta) - \mathcal{J}_I(\rho^1, \theta)\right| \le CT\sup_{t\in[0,T]}\mathbf{d}_1(\rho^1(t), \rho^2(t))\|\mathbf{s}_\theta\|^2_{C^2(\Omega\times[0,T])} \le CT\mathbf{d}_1(\pi^1, \pi^2)\|\mathbf{s}_\theta\|^2_{C^2(\Omega\times[0,T])}.$$

**Lemma 3** (Lemma 8.3 in (Mimikos-Stamatopoulos et al., 2024)). *Let $\pi^\epsilon = \pi \star \Gamma_\epsilon$, and $\pi_G^{N,\epsilon}$ be as in Theorem 1 with $\epsilon < 1$. There exists a dimensional constant $C = C(d) > 0$ such that*

$$\mathbf{d}_1(\pi_G^{N,\epsilon}, \pi^\epsilon) \le \mathbf{d}_1(\pi_G^N, \pi), \tag{37}$$

$$\left\|\pi_G^{N,\epsilon} - \pi^\epsilon\right\|_1 \le C\frac{\mathbf{d}_1(\pi_G^N, \pi)}{\sqrt{\epsilon}}, \tag{38}$$

*and*

$$\left\|\pi^\epsilon\log(\pi^\epsilon) - \pi_G^{N,\epsilon}\log(\pi_G^{N,\epsilon})\right\|_1 \le C\left(1 - \frac{d}{2}\log(\epsilon)\right)\frac{\mathbf{d}_1(\pi_G^N, \pi)}{\sqrt{\epsilon}}. \tag{39}$$

*Moreover, let $\eta_G^{N,\epsilon}$ and $\eta^\epsilon$ be solutions to Eq. (36) with initial conditions $\pi_G^{N,\epsilon}$ and $\pi^\epsilon$ respectively. Then for large enough $T$ that depends on $R$ and the dimension $d$ but is independent of random samples or $N$, we have*

$$\int_\Omega \log(\eta_G^{N,\epsilon}(T))\eta_G^{N,\epsilon}(T) - \log(\eta^{\pi,\epsilon}(T))\eta^{\pi,\epsilon}(T)\,\mathrm{d}x \le \frac{C}{\sqrt{T}}\mathbf{d}_1(\pi, \pi_G^N). \tag{40}$$

*Proof.* Inequalities $(37) - (39)$ follow directly from the proof of Lemma 8.3 in (Mimikos-Stamatopoulos et al., 2024). For the bound in Eq. (40), by the convexity of the function $f(x) = x\log x$, we have

$$\int \log(\eta_G^{N,\epsilon}(T))\eta_G^{N,\epsilon}(T) - \eta^{\pi,\epsilon}(T)\log(\eta^{\pi,\epsilon}(T))\,\mathrm{d}x \le \int\left(1 + \log(\eta_G^{N,\epsilon}(T))\right)\mathrm{d}(\eta_G^{N,\epsilon}(T) - \eta^{\pi,\epsilon}(T))$$

$$\le \left\|1 + \log(\eta_G^{N,\epsilon}(T))\right\|_\infty\left\|\eta_G^{N,\epsilon}(T) - \eta^{\pi,\epsilon}(T)\right\|_1.$$

From the proof of Lemma 8.3 in (Mimikos-Stamatopoulos et al., 2024), we have

$$\left\|\eta_G^{N,\epsilon}(T) - \eta^{\pi,\epsilon}(T)\right\|_1 \le \frac{C}{\sqrt{T}}\mathbf{d}_1(\pi_G^{N,\epsilon}, \pi^\epsilon) \le \frac{C}{\sqrt{T}}\mathbf{d}_1(\pi_G^N, \pi),$$

where $C > 0$ is a dimensional constant. It remains to bound $\left\|1 + \log(\eta_G^{N,\epsilon}(T))\right\|_\infty$. Indeed, by the property of the heat kernel on $R\mathbb{T}^d$, $\eta^{N,\epsilon}(t) \lesssim_{d,R} 1 + (\epsilon + T)^{-d/2}$, and it is lower bounded by $\eta^{N,\epsilon}(t) \gtrsim_{d,R} (\epsilon + T)^{-d/2}$. By Proposition 3, we have $\inf_{x\in\Omega}\eta^{N,\epsilon}(x,t) \le \eta_G^{N,\epsilon}(x,t) \le \sup_{x\in\Omega}\eta^{N,\epsilon}(x,t)$ for any $t$. This finishes the proof. □

*Proof of Eq. (35).* Note that $\mathcal{J}_D(\eta^{\pi,\epsilon}, \theta) = \mathcal{J}_I(\eta^{\pi,\epsilon}, \theta) + 4\|\nabla\sqrt{\eta^{\pi,\epsilon}}\|_2^2$. We have

$$\mathcal{J}_D(\eta^{\pi,\epsilon}, \theta) = \mathcal{J}_D(\eta_G^{N,\epsilon}, \theta) + 4\left(\|\nabla\sqrt{\eta^{\pi,\epsilon}}\|_2^2 - \left\|\nabla\sqrt{\eta_G^{N,\epsilon}}\right\|_2^2\right) + \left(\mathcal{J}_I(\eta^{\pi,\epsilon}, \theta) - \mathcal{J}_I(\eta_G^{N,\epsilon}, \theta)\right).$$

By assumption we have $\mathcal{J}_D(\eta_G^{N,\epsilon}, \theta) \le e_{nn}$. By Lemma 1, we have

$$\|\nabla\sqrt{\eta^{\pi,\epsilon}}\|_2^2 - \left\|\nabla\sqrt{\eta_G^{N,\epsilon}}\right\|_2^2$$

$$= \int_\Omega \pi^\epsilon \log(\pi^\epsilon) - \pi_G^{N,\epsilon}\log(\pi_G^{N,\epsilon})\,\mathrm{d}x + \int_\Omega \eta_G^{N,\epsilon}(T)\log(\eta_G^{N,\epsilon}(T)) - \eta^{\pi,\epsilon}(T)\log(\eta^{\pi,\epsilon}(T))\,\mathrm{d}x.$$

From Eq. (39) in Lemma 3, we can bound the first integral; while the second integral can be bound by Eq. (40). Combining with Lemma 2, we finish the proof. $\qquad\square$

*Proof of Corollary 1.* Note that $\mathcal{M}$ is compact and can be covered by finitely many charts, where the map in each chart is Lipschitz (though with possibly different Lipschitz constant within each chart), so $\mathcal{M}$ has a Riemannian metric that is equivalent to the Euclidean metric in the ambient space. Hence we can apply the result in (Tahmasebi & Jegelka, 2024) to $\mathbb{E}[\mathbf{d}_1(\pi_G^N, \pi)]$. $\qquad\square$

## B PROOF THAT SCORE-BASED GENERATIVE MODELS ARE INTRINSICALLY EQUIVARIANT

*Proof of Theorem 5.* From (Zhang & Katsoulakis, 2023), it is known that score-based generative models are the solution of a mean-field game

$$\begin{cases} \partial_t \rho + \nabla\cdot(\rho(f - \sigma^2\nabla U)) = \dfrac{\sigma^2}{2}\Delta\rho \\ -\partial_t U - f^\top\nabla U + \dfrac{1}{2}|\sigma\nabla U|^2 + \nabla\cdot f = \dfrac{\sigma^2}{2}\Delta U \\ U(x,T) = -\log\pi(x),\ \rho(x,0) = e^{-U(x,0)}. \end{cases} \tag{41}$$

Let $G$ be some group, $g \in G$ be an element of the group, and $A_g$ be the group action corresponding with $g$. Assume data distribution $\pi$ is $G$-invariant Then it is clear that $U(x,T)$ is also $G$-invariant as

$$U(gx,T) = -\log\pi(gx,T) = -\log\pi(x,t) = U(x,T). \tag{42}$$

Furthermore, since $f$ is assumed to be $G$-equivariant, the corresponding Hamilton-Jacobi-Bellman equations are identical for all $g \in G$. Therefore, by the uniqueness of the solution to the Hamilton-Jacobi-Bellman equation, $U(gx,t) = U(x,t)$ for all $t \in [0,T]$. For the existence and uniqueness of smooth solutions of the HJB equation and their properties we refer to (Tran, 2021) (Section 1.7 and references therein), see also (Fleming & Soner, 2006). Therefore, the solution of the HJB equation $U(x,t)$ is invariant, and therefore the score function $\mathbf{s} = -\nabla U$ must be $G$-equivariant. Moreover, it is shown in (Zhang & Katsoulakis, 2023) that the minimizer of the implicit score matching objective, and therefore the ESM, is equivalent to the solution of 41. Therefore, this shows that the neural net must be parameterized in a way that is $G$-equivariant, thus incorporating an induced, equivariant (structural) inductive bias. $\qquad\square$

## C PROOF OF PROPOSITIONS OF VECTOR FIELDS

$G$-**equivariance of** $S_G^E[\mathbf{s}]$. For any $\bar{g} \in G$, we have

$$S_G^E[\mathbf{s}](\bar{g}x,t) = \int_G A_g^\top \cdot \mathbf{s}(g\bar{g}x,t)\mu_G(\mathrm{d}g)$$

$$= \int_G A_{\bar{g}}A_{\bar{g}}^\top A_g^\top \cdot \mathbf{s}(g\bar{g}x,t)\mu_G(\mathrm{d}g)$$

$$= \int_G A_{\bar{g}}A_{g\circ\bar{g}}^\top \cdot \mathbf{s}(g\bar{g}x,t)\mu_G(\mathrm{d}g)$$

$$= A_{\bar{g}}S_G^E[\mathbf{s}](x,t).$$

*Proof of Theorem 3.* It is sufficient to look at the integration of $x$ over $\Omega$. We have

$$\int_\Omega \left( |\mathbf{s}|^2 + 2\nabla \cdot \mathbf{s} \right) S^G[\rho](x)\,\mathrm{d}x = \int_\Omega S_G \left[ |\mathbf{s}|^2 + 2\nabla \cdot \mathbf{s} \right] \rho(x)\,\mathrm{d}x$$

$$= \int_\Omega |\mathbf{s}|^2 \, \rho(x)\,\mathrm{d}x + 2 \int_\Omega S_G \left[ \nabla \cdot \mathbf{s} \right] \rho(x)\,\mathrm{d}x,$$

where the last equality is due to that the module $|\mathbf{s}|$ is $G$-invariant since $\mathbf{s}$ is $G$-equivariant. For the second integral, we have

$$\int_\Omega S_G \left[ \nabla \cdot \mathbf{s} \right] \rho(x)\,\mathrm{d}x = \int_\Omega \int_G \sum_{i=1}^d \frac{\partial(\mathbf{s}_i)}{\partial x_i}(gx)\,\mathrm{d}\mu_G(g)\rho(x)\,\mathrm{d}x$$

$$= \int_G \int_\Omega \sum_{i=1}^d \frac{\partial(\mathbf{s}_i)}{\partial x_i}(gx)\rho(x)\,\mathrm{d}(x)\,\mathrm{d}\mu_G(g)$$

$$= \int_G \int_\Omega \sum_{i=1}^d \frac{\partial(\mathbf{s}_i)}{\partial x_i}(x)\rho(g^{-1}x)\,\mathrm{d}(g^{-1}x)\,\mathrm{d}\mu_G(g)$$

$$= -\int_G \int_\Omega \mathbf{s}(x)^\top (A_g \nabla\rho|_{g^{-1}x})\,\mathrm{d}(g^{-1}x)\,\mathrm{d}\mu_G(g) \quad \text{(use integration by parts)}$$

$$= -\int_G \int_\Omega (A_g^\top \mathbf{s}(x))^\top (\nabla\rho|_{g^{-1}x})\,\mathrm{d}(g^{-1}x)\,\mathrm{d}\mu_G(g)$$

$$= -\int_G \int_\Omega (\mathbf{s}(g^{-1}x))^\top (\nabla\rho|_{g^{-1}x})\,\mathrm{d}(g^{-1}x)\,\mathrm{d}\mu_G(g) \quad \text{(by the equivariance of } \mathbf{s})$$

$$= -\int_G \int_\Omega (\mathbf{s}(x))^\top (\nabla\rho(x))\,\mathrm{d}x\,\mathrm{d}\mu_G(g)$$

$$= \int_G \int_\Omega (\nabla \cdot \mathbf{s})(x)\rho(x)\,\mathrm{d}x\,\mathrm{d}\mu_G(g)$$

$$= \int_\Omega (\nabla \cdot \mathbf{s})(x)\rho(x)\,\mathrm{d}x.$$

Therefore, we have

$$\int_\Omega \left( |\mathbf{s}|^2 + 2\nabla \cdot \mathbf{s} \right) S^G[\rho](x)\,\mathrm{d}x = \int_\Omega \left( |\mathbf{s}|^2 + 2\nabla \cdot \mathbf{s} \right) \rho(x)\,\mathrm{d}x.$$

$\square$

To prove Proposition 1, we need the following lemma.

**Lemma 4.** *For a generic $\rho \in \mathcal{P}(\Omega)$, which may not be $G$-invariant, the score formula of its symmetrized measure $S^G[\rho]$, is given by*

$$\nabla_x \left[ \log \left( S^G[\rho] \right) \right](x) = \frac{\int_G A_g^\top \cdot (\nabla_x \rho)|_{gx}\,\mathrm{d}\mu_G(g)}{\int_G \rho(gx)\,\mathrm{d}\mu_G(g)},$$

*where $(\nabla_x \rho)|_{gx}$ is the gradient of $\rho$ evaluated at $gx$.*

*Proof of Lemma 4.*

$$\nabla_x \left[ \log \left( S^G[\rho] \right) \right](x) = \nabla_x \left[ \log \left( \int_G \rho(gx)\,\mathrm{d}\mu_G(g) \right) \right]$$

$$= \frac{\nabla_x \int_G \rho(gx)\,\mathrm{d}\mu_G(g)}{\int_G \rho(gx)\,\mathrm{d}\mu_G(g)}$$

$$= \frac{\int_G \nabla_x \rho(gx)\,\mathrm{d}\mu_G(g)}{\int_G \rho(gx)\,\mathrm{d}\mu_G(g)}$$

$$= \frac{\int_G A_g^\top \cdot (\nabla_x \rho)|_{gx} \, \mathrm{d}\mu_G(g)}{\int_G \rho(gx) \, \mathrm{d}\mu_G(g)}.$$

$\square$

*Proof of Proposition 1.* It suffices to prove the result for each time $t$, so we omit the time parameter. Let $\Omega/G$ be the quotient space of $\Omega$ by $G$. By the definition in Eq. (8), denoting by $\nabla \log \rho|_{gx}$ the score $\nabla \log \rho$ evaluated at $gx$, up to a multiplicative constant $C_G$ the depends on $G$ ($C_G = 1$ if $\dim(\Omega/G) < d$ and $C_G = |G|$ if $G$ is finite), we have

$$\mathcal{J}_E(\rho, \mathbf{s}) = C_G \int_{\Omega/G} \int_G \left| \mathbf{s}(gx) - \nabla \log \rho|_{gx} \right|^2 \rho(gx) \, \mathrm{d}\mu_G(g) \, \mathrm{d}x$$

$$= C_G \int_{\Omega/G} \int_G \left| A_g \cdot \mathbf{s}(x) - \nabla \log \rho|_{gx} \right|^2 \rho(gx) \, \mathrm{d}\mu_G(g) \, \mathrm{d}x$$

$$= C_G \int_{\Omega/G} \int_G \left| \mathbf{s}(x) - A_g^\top \cdot \nabla \log \rho|_{gx} \right|^2 \rho(gx) \, \mathrm{d}\mu_G(g) \, \mathrm{d}x,$$

where the last equality is due to the group actions in $G$ are isometries. For each $x \in \Omega/G$, regardless of $C_G$, we have

$$\nabla_\mathbf{s} \left[ \int_G \left| \mathbf{s}(x) - A_g^\top \cdot \nabla \log \rho|_{gx} \right|^2 \rho(gx) \, \mathrm{d}\mu_G(g) \right] = 2 \int_G \mathbf{s}(x) - A_g^\top \cdot (\nabla \log \rho|_{gx}) \rho(gx) \, \mathrm{d}\mu_G(g).$$

Then the stationary point of the above equation is given by

$$\mathbf{s}^*(x) = \frac{\int_G A_g^\top \cdot (\nabla \log \rho|_{gx}) \rho(gx) \, \mathrm{d}\mu_G(g)}{\int_G \rho(gx) \, \mathrm{d}\mu_G(g)}.$$

Note that $\nabla \log \rho|_{gx} = \frac{(\nabla_x \rho)|_{gx}}{\rho(gx)}$. This combined with Lemma 4 proves the claim. $\square$

*Proof of Theorem 4.* It suffices to prove the equality for each time $t$, thus we will omit the time parameter. Expanding the square, it is equivalent to show that

$$\int_\Omega (\mathbf{s}^\top \nabla \log \rho) \rho(x) \, \mathrm{d}x = \int_\Omega \left( \mathbf{s}^\top S_G^E[\mathbf{s}] - \left| S_G^E[\mathbf{s}] \right|^2 + S_G^E[\mathbf{s}]^\top \nabla \log \rho \right) \rho(x) \, \mathrm{d}x.$$

First, we show that $\int \mathbf{s}^\top S_G^E[\mathbf{s}] \rho(x) \, \mathrm{d}x = \int \left| S_G^E[\mathbf{s}] \right|^2 \rho(x) \, \mathrm{d}x$. We have

$$\text{LHS} = \int_\Omega \int_G \mathbf{s}(x)^\top \cdot A_g^\top \mathbf{s}(gx) \, \mathrm{d}\mu_G(g) \rho(x) \, \mathrm{d}x$$

by the definition of the operator $S_G^E$; while

$$\text{RHS} = \int_\Omega \int_G \int_G \mathbf{s}(g_1 x)^\top A_{g_1} A_{g_2}^\top \mathbf{s}(g_2 x) \, \mathrm{d}\mu_G(g_1) \, \mathrm{d}\mu_G(g_2) \rho(x) \, \mathrm{d}x$$

$$= \int_\Omega \int_G \int_G \mathbf{s}(g_1 x)^\top A_{g_2 \circ g_1^{-1}}^\top \mathbf{s}(g_2 x) \, \mathrm{d}\mu_G(g_1) \, \mathrm{d}\mu_G(g_2) \rho(x) \, \mathrm{d}x$$

$$= \int_G \int_G \int_\Omega \mathbf{s}(g_1 x)^\top A_{g_2 \circ g_1^{-1}}^\top \mathbf{s}(g_2 x) \rho(x) \, \mathrm{d}x \, \mathrm{d}\mu_G(g_1) \, \mathrm{d}\mu_G(g_2)$$

$$= \int_G \int_G \int_\Omega \mathbf{s}(x)^\top A_{g_2 \circ g_1^{-1}}^\top \mathbf{s}(g_2 \circ g_1^{-1} x) \rho(x) \, \mathrm{d}x \, \mathrm{d}\mu_G(g_1) \, \mathrm{d}\mu_G(g_2)$$

$$= \int_G \int_G \int_\Omega \mathbf{s}(x)^\top A_g^\top \mathbf{s}(gx) \rho(x) \, \mathrm{d}x \, \mathrm{d}\mu_G(g) \, \mathrm{d}\mu_G(g_2)$$

$$= \int_G \int_\Omega \mathbf{s}(x)^\top A_g^\top \mathbf{s}(gx) \rho(x) \, \mathrm{d}x \, \mathrm{d}\mu_G(g) = \text{LHS}$$

where the fourth equality is due to the $G$-invariance of $\rho$ and $A_g$ is unitary for any $g \in G$, and the fifth equality is due to that $G$ is unimodular so the Haar measure $\mathrm{d}_{\mu_G}$ is left-, right- and inverse-invariant.

Then it remains to show that $\int (\mathbf{s}^\top \nabla \log \rho)\rho(x)\,\mathrm{d}x = \int (S_G^E[\mathbf{s}]^\top \nabla \log \rho)\rho(x)\,\mathrm{d}x$. Indeed, we have

$$
\begin{aligned}
\int_\Omega (S_G^E[\mathbf{s}]^\top \nabla \log \rho)\rho(x)\,\mathrm{d}x &= \int_\Omega \int_G (A_g^\top \mathbf{s}(gx))^\top \,\mathrm{d}\mu_G(g)(\nabla \log \rho(x))\rho(x)\,\mathrm{d}x \\
&= \int_G \int_\Omega \mathbf{s}(gx)^\top A_g (\nabla \log \rho(x))\rho(x)\,\mathrm{d}x\,\mathrm{d}\mu_G(g) \\
&= \int_G \int_\Omega \mathbf{s}(gx)^\top (\nabla \log \rho|_{gx})\rho(x)\,\mathrm{d}x\,\mathrm{d}\mu_G(g) \\
&= \int_G \int_\Omega \mathbf{s}(x)^\top (\nabla \log \rho(x))\rho(x)\,\mathrm{d}x\,\mathrm{d}\mu_G(g) \\
&= \int_\Omega \mathbf{s}(x)^\top (\nabla \log \rho(x))\rho(x)\,\mathrm{d}x,
\end{aligned}
$$

where the 3-rd equality is due to that $\nabla \log \rho$ is $G$-equivariant, and the 4-th equality is by a change of variable and $\rho$ is $G$-invariant. $\qquad\square$

*Proof of Proposition 2.* Let $\Gamma = \mathrm{Lip}_1(\Omega)$, and $\Gamma_G^{inv}$ be the subspace of $\Gamma$ that consists of $G$-invariant functions. By Assumption 1, actions in $G$ are 1-Lipschitz. Thus, $S_G[\Gamma] \subset \Gamma$. First note that $S^G[\pi] = \pi$ since $\pi$ is $G$-invariant. Then we have

$$
\begin{aligned}
\mathbf{d}_1(S^G[\eta], \pi) &= \mathbf{d}_1(S^G[\eta], S^G[\pi]) \\
&= \sup_{\gamma \in \Gamma} \left\{ \mathbb{E}_{S^G[\eta]}[\gamma] - \mathbb{E}_{S^G[\pi]}[\gamma] \right\} \\
&= \sup_{\gamma \in \Gamma_G^{inv}} \left\{ \mathbb{E}_\eta[\gamma] - \mathbb{E}_\pi[\gamma] \right\} \\
&\leq \sup_{\gamma \in \Gamma} \left\{ \mathbb{E}_\eta[\gamma] - \mathbb{E}_\pi[\gamma] \right\} = \mathbf{d}_1(\eta, \pi),
\end{aligned}
$$

where the second equality is by the definition of $\mathbf{d}_1$ metric, and the third equality is due to Theorem 4.6 in (Birrell et al., 2022). $\qquad\square$

