# OpenReview forum: "Equivariant score-based generative models provably learn distributions with symmetries efficiently"
_ICLR.cc/2025/Conference — Submitted to ICLR 2025_

### Official Review · Reviewer_cTjt · 2024-10-29

**Soundness:** 3
**Presentation:** 3
**Contribution:** 2
**Rating:** 5
**Confidence:** 4

**Summary:**

> This paper addresses how generative models can benefit from leveraging symmetries in data distributions to improve sampling efficiency and generalization on score-based generative models (SGMs). The authors focus on the empirical success of SGMs in scenarios where data has intrinsic symmetries (e.g., physics, molecular simulations). They argue that symmetries embedded in data distributions, known as group symmetries, can lead to improved model performance.

> First, they analyze how equivariant score-matching (with vector fields preserving symmetry) is an optimal method for training SGMs with symmetric data, bypassing the need for extensive data augmentation. Building on *Mimikos-Stamatopoulos et al. (2024)*, the authors generalize their bound errors in SGMs, breaking it down into errors stemming from non-equivariance, score-matching, sample complexity, and early stopping. They use the Wasserstein-1 distance (*d1*) as a metric to show how errors in these areas impact model performance, relying on HJB techniques and the mFG interpretation of SGMs.
The authors provides also numerical simulations aiming to corroborate their theoretical claims and to illustrate that embedding equivariant structures directly into SGMs consistently outperforms data-augmented models in generating high-quality samples of symmetric distributions, with lower Wasserstein-1 distance errors.

> This paper advances the theoretical understanding of symmetry in generative models, offering both conceptual and practical insights into how SGMs can be designed to better handle symmetric data distributions, efficiently using equivariant inductive biases instead of data augmentation.

**Strengths:**

The article demonstrates several notable strengths that advance the understanding and effectiveness of score-based generative models by focusing on equivariance:
- **Efficient Generalization of Existing Literature:** Building on the recent work by *Mimikos-Stamatopoulos et al. (2024)*, this article rigorously shows that $G$-invariance (group symmetry) is a key property that strengthens the structure and efficiency of SGMs. By extending these theoretical foundations, the authors highlight equivariance as crucial for improving model convergence and generalization.
- **Insightful Theoretical Framework Linking Key Concepts:** The paper effectively utilizes connections between MFG optimization, the HJB equation, and SGMs to clarify the structural properties of these models. This theoretical linkage reveals how SGMs can leverage inherent symmetries, providing a clearer perspective on the convergence bounds and making the role of symmetry more explicit.
- **Detailed Separation of Symmetry Components in Convergence Bounds:** The authors efficiently disentangle the components of symmetry within convergence bounds, offering a nuanced view of how these factors contribute to model stability and performance. This separation facilitates a better understanding of the role of symmetry in error decomposition and contributes to more accurate theoretical predictions.
- **Guidance for Model Enhancement:** The article provides practical suggestions for leveraging $G$-invariance directly within model architectures. This approach minimizes the need for data augmentation and improves computational efficiency while supporting more robust and generalizable models, laying groundwork for future research that optimally exploits symmetry.

**Weaknesses:**

- **Bounds Inefficiency:** The generalization bounds presented appear to lack efficiency, especially as they depend on the radius $R$ of the compact domain in which the data is confined. This leads to a bound that increases with the data’s spatial constraints. Additionally, the time horizon $T$ dependency in the bound further amplifies this issue, potentially limiting the practical applicability of the results.
- **Limited Novelty in Symmetry Component Analysis:** While the authors successfully separate the symmetry component in the bounds from *Mimikos-Stamatopoulos et al. (2024)*, the originality of this analysis seems limited. Revisiting classical results without substantial improvements in simulation outcomes weakens the impact of this contribution on advancing symmetry-related insights.
- **Insufficient Context for Symmetry in Examples:** Although the paper addresses how the absence of symmetry could affect convergence bounds, it lacks clear examples that naturally demonstrate symmetry or its absence. A more thorough description of practical cases where symmetry is essential would strengthen the authors’ arguments and provide clearer motivation.

**Questions:**

- **Limited Scope of Numerical Experiments:** The numerical experiments are minimal and fail to comprehensively illustrate the necessity of incorporating equivariance into the SGM architecture. The four-Gaussian example appears overly simplistic, raising questions about the generality of the findings. Moreover, even in this simple example the benefit of using equivariance adjustments does not seem to clear. Could a more complex real-world example better justify the proposed framework? Also, is the bound evaluated and, if so, is it tight?
- **Choice of Architecture:** Why did the authors choose the current architecture over U-Net, which has been crucial to the success of SGMs due to its computational efficiency and architectural adaptability? Clarifying the rationale could provide insights into the model’s design choices.
- **Inconsistencies Between Theory and Experiments:** The authors constrain the diffusion process to a compact set but then use a Gaussian distribution in experiments, which does not align with this assumption. How should the bound be applied or evaluated in this case? Addressing this discrepancy would clarify the applicability of the results across different setups.

---

> ### Author Response · Authors · 2024-11-20
> **Author Response**
>
> **Weaknesses**
>
> 1. We consider the domain to be bounded, with $R$ fixed, and it should not be taken to $\infty$. As noted in our Conclusion and Future Work, applying the Ornstein–Uhlenbeck (OU) process to unbounded domains would be a more appropriate approach in such cases. Furthermore, in practice, $T$ never approaches $\infty$. Instead, it functions as a multiplicative factor in the sample complexity and does not constrain practical applicability.
>
>
> 2. We want to remind the reviewer that our contributions to Symmetry Component Analysis in Section 4 are new and independent of Mimikos-Stamatopoulos et al. (2024). Furthermore,  Mimikos-Stamatopoulos et al. (2024) is not ``classical" but rather recent. We provide, for the first time, the quantification of errors in score-based generative modeling due to the approximation of equivariant score functions with non-equivariant vector fields.
>
>
> 3. We have provided abundant references in the Introduction and Related Works for the importance the motivation of encoding symmetries into generative models.
>
> **Questions**
>
> 1. We refer the reviewer to the recent paper (Lu et al., 2024) cited in our paper for extensive experiments and implementations for equivariant SGMs, which also directly inspired our current *theoretical* study. We emphasize, however, that our primary contribution is to *explain* the existing widely observed empirical results and to provide theoretical justification for the use of equivariant models in contrast to data augmentation.
>
>
> 2. The design or the architecture of the model is not the focus of our work, and the parametrization of equivariant neural networks is not unique. The main contribution of our work is to quantitatively compare the role of equivariant models with that of data augmentation, and our simple simulation using fully-connected ReLU networks also provides a fair comparison. We also refer the reviewer to (Lu et al., 2024) for a few experiments of equivariant SGMs using U-Nets.
>
>
> 3. We assume the domain to be bounded for technical simplicity to provide the first quantitative comparison between model equivariance and data augmentation. Although the Gaussian example has unbounded support, it supports that our conclusion to advocate model equivariance is not limited to bounded domains. But for the theoretical part, we should use the OU process instead as mentioned in our future work and conclusion section.
>
> ---
> Finally, the review has some vague language and inconsistencies, making it difficult to properly address them. We provide you with some examples.
>
> * "Insightful Theoretical Framework Linking Key Concepts: This theoretical linkage reveals how SGMs can leverage inherent symmetries, providing a clearer perspective on the convergence bounds and making the role of symmetry more explicit.'' Our HJB theorem (Theorem 3) is different from the result for the convergence bounds (Theorem 1,2).
>
> * "Bounds Inefficiency'' seems to us vague and appears to ignore our assumptions on bounded domains. As we assume the domain to be bounded, the bound should depend on $R$. Therefore, it does not make sense to take $R$ to infinity under our assumption: one should use the OU process instead, as mentioned in our future work section.
>
> * "Choice of Architecture: Why did the authors choose the current architecture over U-Net, which has been crucial to the success of SGMs due to its computational efficiency and architectural adaptability? Clarifying the rationale could provide insights into the model’s design choices." We do not use anywhere image-based data sets hence there is no need to consider U-nets or other specialized architectures. The problem we study is not about whether we should use U-Net or not, but rather whether we should use equivariant score approximations.
>
> * "While the authors successfully separate the symmetry component in the bounds from Mimikos-Stamatopoulos et al. (2024), the originality of this analysis seems limited. Revisiting classical results without substantial improvements in simulation outcomes weakens the impact of this contribution on advancing symmetry-related insights." The paper Mimikos-Stamatopoulos et al. (2024) was just accepted at NeurIPS 2024 and is not quite yet a ``classic".
>
> * The section "Detailed Separation of Symmetry Components in Convergence Bounds" in the Strengths seems to be somewhat contradictory to the section called  ``Limited Novelty in Symmetry Component Analysis'' in the Weaknesses.
>
> * The other Weakness, namely  "Insufficient Context for Symmetry in Examples" appears to be rather vague and does not provide actionable items for us to address, given the already extensive discussion of the bibliography on related questions in the manuscript. We also refer the reviewer to our Global Response.

---

> > ### Comment · Reviewer_cTjt · 2024-11-24
> >
> > We thank the authors for their efforts in addressing the questions raised during the review process. Below, I provide my evaluation of the submission:
> > 1. While "paper Mimikos-Stamatopoulos et al. (2024) was just accepted at NeurIPS 2024 and is not quite yet a ``classic" ", it corresponds to a contribution external to the paper under analysis. This paper builds heavily on that prior work, which limits the originality of the theoretical insights presented here. While the submission demonstrates some interesting theoretical analyses, the modifications introduced appear relatively modest. In my view, the scientific innovation presented may not yet align with the high bar typically expected for a venue of ICLR’s caliber.
> > 2. The submission’s analysis relies on a paper explicitly acknowledged as “not yet classic,” which raises questions about the broader relevance of the findings in light of the existing extensive literature on convergence bounds. Specifically: (a) The numerical experiments involve examples that do not fully align with the assumptions required for the application of the proposed bounds. For instance, the radius $R$ cannot be extended to  $+\infty$ , and no potential truncation errors are accounted for in the analysis. (b) The tightness and applicability of the proposed bound for $G$-invariant distributions have not been fully substantiated, either theoretically or empirically. A more thorough evaluation of the bound’s relevance, either through numerical experiments or theoretical validation on appropriate examples, would significantly strengthen the impact of this work.
> >
> > I appreciate the authors’ dedication and effort in developing this submission, but for the reasons outlined above, I maintain my current score on this submission.

---

### Official Review · Reviewer_eH4X · 2024-11-01

**Soundness:** 3
**Presentation:** 4
**Contribution:** 3
**Rating:** 6
**Confidence:** 3

**Summary:**

For learning distributions with certain symmetries using score-based generative models, this paper performs the first theoretical analysis and provides some guarantees. In particular, the authors show an improved error bound when the data distribution is group-invariant, use Hamilton-Jacobi-Bellman (HJB) theory to describe the inductive bias, and provide numerical tests to reinforce their claims.

**Strengths:**

The paper provides many theoretical results in a rigorous mathematical setting, when the target distribution has some symmetries. Furthermore, many of the results directly translate to practical use cases (e.g., Theorem 4).

**Weaknesses:**

While the (only) numerical example well demonstrates the theoretical results, it is a very simple one-dimensional experiment. It would have been much more convincing to see the theory validated with a higher-dimensional learning problem. For example, there are several high-dimensional experiments in Birrell et al. (2022) in ICML (cited by the present paper) and I wonder if the authors can validate their theoretical results in any of them.

**Questions:**

1. I understand that G is a given group when it comes to the theoretical results, but what exactly is the G group when it comes to the numerical results? For instance, in Section 6, is the group G the collection of four Gaussians or some properties of these four Gaussians (or other things)? How the elements of the example correspond to the elements of the theoretical parts need to be addressed in more detail. Specifically, I wonder if the authors can explicitly define the G group for the numerical example and provide a clear mapping between the theoretical elements and their counterparts in the experiment.

2. In the Appendix, at lines 748-752, I do not understand how applying Theorem 6 yields the approximate inequality. If you already have the exact inequality in Theorem 6, why don't we just use the exact inequality (instead of some approximation)? I wonder if the authors can provide a step-by-step derivation of how they arrive at the approximate inequality from Theorem 6, or explain why an approximation is used if an exact inequality is available.

---

> ### Author Response · Authors · 2024-11-20
> **Author Response**
>
> We thank the reviewer for the feedback and for carefully reading our manuscript. Please see our responses below.
>
> **Weaknesses**
>
> We did not conduct extensive experiments, as numerous related experiments have already been presented in (Lu et al., 2024) cited in our paper. Lu et al., 2024 has several similar experiments as in Birrell et al. (2022) in ICML, such as rotated MNIST, LYSTO, and ANHIR. Similar observations have also been made in earlier works on GANs and normalizing flows, where the generator/discriminator in GANs is equivariant/invariant, and the velocity field in CNFs is equivariant. However, common in all these approach is that they lack a theory to explain why model equivariance is more essential than data augmentation. One of our main contributions is to provide the first quantitative explanation for why model equivariance is more critical than simply performing data augmentation, in the context of SGMs.
>
>
> **Questions**
>
> 1. The group used in the numerical experiments is $C_4$, the rotational group, represented by rotation matrices of $0^\circ$, $90^\circ$, $180^\circ$, and $270^\circ$. Clearly, the mixture of four Gaussians is invariant for any combination of these rotations. To be specific, the representations of the group actions are
> \begin{align*}
>     A_0 = \begin{bmatrix}
>        1 & 0 \\\ 0 & 1
>     \end{bmatrix},  A_{90} = \begin{bmatrix}
>        0 & -1 \\\ 1 & 0
>     \end{bmatrix}, A_{180} = \begin{bmatrix}
>        -1 & 0 \\\ 0 & -1
>     \end{bmatrix}, A_{270} = \begin{bmatrix}
>        0 & 1 \\\ -1 & 0
>     \end{bmatrix}
> \end{align*}
>
>
> 2. We assume the reviewer refers to the symbol '$\lesssim$' at Line 751. If that is the case, it is a notation, and we use it to mean up to a dimensional constant $C=C(d)>0$ in Theorem 6, and it is equivalent to rewriting it as $\leq C$ by putting $C$ back into Line 751, essentially a direct application of Theorem 6. We will add a comment about that in the proof to clarify the notation.

---

> > ### Comment · Reviewer_eH4X · 2024-12-03
> >
> > After reading the authors' responses to other reviewer comments, I agree that a systematic numerical study is outside the scope of the paper. Indeed, equivariant SGMs are well-studied in the paper cited by the authors (Lu, 2024). There is no need to redo their experiments for this work.
> >
> > The purpose of this work is to provide a theoretical study of learning group symmetries with equivariant SGMs, and I believe the work achieved that goal. As a result, I am open to raising the rating to 8.

---

> > > ### Author Response · Authors · 2024-12-04
> > >
> > > Thank you for considering increasing the rating to 8—we appreciate your time and support.

---

### Official Review · Reviewer_Y16Q · 2024-11-03

**Soundness:** 3
**Presentation:** 3
**Contribution:** 3
**Rating:** 6
**Confidence:** 2

**Summary:**

This paper presents the first theoretical analysis of score-based generative models (SGMs) for learning distributions with symmetries. It proves that incorporating symmetries through equivariant score approximations leads to improved generalization bounds and sampling efficiency. The study demonstrates that equivariant SGMs can effectively learn symmetric distributions without needing data augmentation and quantifies the impact of non-equivariant score parametrization on generalization error. The theoretical results are investigated with simple numerical experiments.

**Strengths:**

The paper is easy to read and provides a new theoretical anlysis for equivariant score-based generative models for learning symmetric distributions, providing valuable insights into their improved generalization and sample efficiency. The theoretical results are able to provide practical insights on comparison of equivariance with data augmentation.

The paper presents a first-time quantitative comparison between data augmentation and the use of equivariant score parametrizations, demonstrating the advantages of the latter in terms of generalization error bounds.

**Weaknesses:**

There are multiple weakness of this paper:
Major:
The numerical experiments, are conducted on extremely simple examples and the conclusions reported do not appear to follow directly from the example considered. Score based models are often used in much more complex, higher-dimensional datasets. I expand further on this in the questions sections. Furthermore, the main result of Section 4.1 - Theorem 3 seems completely irrelevant to the rest of the paper (or at least not sufficiently connected to it - happy to be proven wrong).

Minor:
The analysis focuses on symmetries representable by unitary matrices, significantly limiting its applicability. While the paper advocates for equivariant SGMs, it doesn't fully address the potential increase in computational complexity associated with designing and training equivariant neural networks.

**Questions:**

* Line 140: Clasically the projection $S_G$ is referred to as the Reynolds operator.
* In Line (3) - doesnt $\nabla \text{log} \rho$ transform as $A^{\top}\nabla \text{log} \rho(Ax)$?
* Line 296 - it seems that the main point is the improved scaling of $1/d^*$, though not emphasised.
* Theorem 3 appears to be almost completely irrelevant? Could you help me understand why the result is either not obvious or expand on why exactly it is relevant?
* Line 370 onwards I am not sure how the analogy with symplectic integrators fits into the picture there
* I beleive that remark 3 once again is entirely obvious without the theorems from the objective function?
* Line 430 - this seems to completely miss the point of non-optimisability. Oftentimes, optimizing equivariant functions is harder/more expensive. And while $e_nn$ can be made as small as possible in theory, practice will be different.
* Figure 1 - I am confused why blue and orange are not the same exact curves - the loss itself must be identical? Even more surprisingly - why are red and green so close?
* I believe that the approach that you use for approximating the Wasserstein distance is not correct. It is fairly well known in the literature that WGANs significantly overestimate the wasserstein distances: https://arxiv.org/pdf/2103.01678
* Line 491 - I believe that the provided results are not extensive enough to claim neither that training a non-equivariant with augmentation nor equivariant without are superior. What is more, training an equivariant network on augmented data does not appear to make much sense, as the parameter gradient are the same in both cases?

---

> ### Author Response · Authors · 2024-11-20
> **Author Response**
>
> We appreciate the reviewer’s recognition of the strengths of our paper, particularly in acknowledging that 1) it provides the first theoretical analysis of score-based generative models (SGMs) for learning distributions with symmetries, and 2) it offers a first-time quantitative comparison between data augmentation and the use of equivariant score parameterizations.
>
> However, we respectfully disagree with most of the concerns raised, as we elaborate next. A key contribution of our paper lies in the quantitative comparison of building equivariance directly into the model, through equivariant score approximations, versus relying solely on data augmentation. We hope our clarifications below address the reviewer’s concerns and further highlight the significance of our contributions.
>
> **Weaknesses**
>
> 1. We did not conduct extensive experiments, as numerous related experiments have already been presented in (Lu et al., 2024) cited in our paper. Similar observations have also been made in earlier works on GANs and normalizing flows, where the generator/discriminator in GANs is equivariant/invariant, and the velocity field in CNFs is equivariant. Our primary focus is twofold: 1) to provide the first rigorous explanation of why model equivariance is fundamentally more impactful than merely applying data augmentation, specifically in the context of SGMs, and 2) to quantitatively analyze the resulting improvements in the generalization bound. The numerical example included in our work serves to validate the theoretical findings.
>
>
> 2. Theorem 3 rigorously identifies the intrinsic *inductive bias of score-based generative models*, which justifies the importance and necessity of using $G$-equivariant vector fields using the fundamental characterization of SGMs as a mean-field game. This result is new and not provided in previous symmetry-related works. The main focus of our work is to *rigorously justify* the necessity and importance of building symmetry into SGMs, and therefore Theorem 3 is a key result in providing a rigorous foundation for this goal.
>
> 3. The characterization that our analysis only applies to groups represented by unitary matrices is categorically false. We emphasize, again, that Theorem 3 is valid for any group $G$ as it *does not rely on a unitary representation*, and shows that the inductive bias of SGMs states that a $G$-invariant distribution demands a $G$-equivariant vector field. This fact is true for any group $G$. Furthermore, unitary representations are not a limitation, as they encompass important transformations such as rotations and reflections. For the subsequent results, we can assume $G$ is compact, which ensures the existence of a unitary representation by leveraging a change of variables through the Peter–Weyl Theorem.
>
> 4. The computational complexity associated with designing and training equivariant neural networks is beyond the scope of our work. However, we refer the reviewer to Group Equivariant Convolutional Networks (G-CNNs) [Cohen \& Welling, ICML 2016] and Steerable CNNs [Cohen \& Welling, ICLR 2017] to address the concerns regarding the efficiency of equivariant neural networks.
>
>
> **Questions**
>
> 1. Thank you for the related notion. We will mention this in the revision.
>
> 2. Equation (3) is the definition of being $G$-equivariant.
>
> 3. Thank you, we will emphasize the improvement in the convergence rate that depends on the intrinsic dimension. This important improvement is only possible because we are using the $\mathbf{d}_1$ metric in our generalization bounds.
>
> 4. Theorem 3 justifies the importance of using $G$-equivariant vector fields from the mean-field perspective. This result is new and not provided in previous symmetry-related works. As the main focus of our work is to rigorously justify the necessity and importance of building symmetry into the model, we do not believe that Theorem 3 is irrelevant. On the contrary, it provides a rigorous explanation of why the equivariance of the vector field is an intrinsic inductive bias, and therefore should be included in the algorithms. Theorem 3 is applicable for any group $G$ without any restrictions.
>
> 5. In practice we only have access to finite samples drawn from the unknown distribution, and those finite samples do not preserve the symmetry even if the underlying distribution does have symmetry. Therefore, the practical numerical schemes should incorporate symmetry so that the outcome would respect the symmetry. ``Symplectic numerical integration'' is one of the well-known examples in numerical analysis that has a similar idea that the numerical scheme should have the structure-preserving properties inherited from the continuous problem, such as Hamiltonian systems and energy conservation.

---

> > ### Comment · Reviewer_Y16Q · 2024-11-20
> >
> > I would like to thank the authors for their responses and clarifications - I completely understand and agree that a full experimental investigation falls outside the scope of this paper. Moreover, I do believe that the paper itself is a significant theoretical contribution - and my initial inclination was to accept the paper. However, there were a number of unsubstantiated claims which led me to lower the score, which I explained above.
> >
> > I do however feel that most of my concerns have been left unanswered. To be precise:
> > *W1: you mention that your work provides a "quantitative comparison of building equivariance directly into the model, through equivariant score approximations, versus relying solely on data augmentation" - however that I fail to see directly how that is (i presume also you mean theoretically?). I do see that Theorem 5 eqn 19 or eqn 21 shows that one can split the error into DFE + JD, and thus a model trained on augmented data is provably worse than the same model trained on augmented data, which is then averaged over the group. However, I do not see, why I would not be able to come up with a non-equivariant model which would make DFE+JD smaller overall. Thus, if the claim were that reynolds averaging always makes your models better - I would completely agree. However the paper currently appears to be claiming more.
> >
> > *W2: I think I understand this bit better now - however I do believe the presentation can significantly be improved. As it currently stands section 4.1 is *completely* separated from the rest of the paper. The connection of MFG to the initial problem in section 2 is not very clear, especially for me, as someone not familiar with this connection. It seems no subsequent results need it or use the MFG formulation. I believe establishing clearer connections to the rest of the paper would be very beneficial.
> >
> > *W3: It seems once again, that section 4.1 was not connected properly to the rest of the paper. As it currently stands, section 4 does not appear to redefine the group - thus, while theorem 3 may work for other groups, Section 2 establishes that in this work Assumption 1 holds that g acts in a linear isometry.
> >
> > I do agree that rotations, translations and reflections are important transformations, it nonnetheless limits the scope of the work, despite being an important first step.
> >
> > It is an interesting point that indeed one can redefine the scalar product, such that the action is unitary - which seems interesting, but it is not immediately clear how the wasserstein-1 distance based results change. Do you have any intuition for this?
> >
> > *W4: Would you mind pointing to the specific results in those papers? To the best of my knowledge they do not directly compare this. While the question of efficacy is outside the scope of the paper, it is directly relevant to the claims, and thus needs to either be mentioned, or the claims need to be clarified.
> >
> > *Q2: That is right, after which you then state that the score is equivariant, but it does not transform according to equation 3. Could you clarify what the disconnect is?
> > *Q4: I suggest you directly state in section 4.1 that theorem 3 does indeed hold for arbitrary groups, without assumption 1.
> > *Q5: I do understand the analogy of building in symmetries, but once again - this analogy seems to be completely misplaced (symplectic integrators turn the problem into a hamiltonian evolution with a shadow hamitlonian, that is completely different from what your work does, if you really want to keep this remark, I would suggest moving it to the introduction).
> >
> > In addition to all of the above, the rest of the questions I have raised appear to be unanswered.

---

> > > ### Author Response · Authors · 2024-11-21
> > > **Response to the comment**
> > >
> > > We thank the reviewer for the prompt response and the engagement in the discussion. Please see our responses to each point below.
> > >
> > > W1: "However, I do not see, why I would not be able to come up with a non-equivariant model which would make DFE+JD smaller overall." -- Your comment indeed includes an important point that our proposed DFE is a measurement of how close your model is to being perfectly $G$-equivariant. Hence if you want to make DFE+JD as small as possible, then you should ensure each of them is small, since both DFE and JD are non-negative. Given both of them are non-negative, this implies DFE should be small, meaning your model is very "close" to being equivariant. "Thus, if the claim were that reynolds averaging always makes your models better - I would completely agree." -- This is absolutely true by our analysis: assuming the model is trained on augmented data in any cases, then you always have a gain on $\mathbb{E}[\mathbf{d}_1(\pi_G^N,\pi)]$ in Eq. (21), and the first two terms in (21) are actually the right-hand side of Eq. (19) in Theorem 5, which says the score-matching error is always smaller using the reynolds averaging $S_G^E[\mathbf{s}]$, and the error reduction is equal to the DFE of the model. Therefore, the overall error is smaller using the reynolds averaging. However, our results are more general, as using the reynolds averaging is not the only way to build equivariant models, for example, G-CNNs [Cohen \& Welling, ICML 2016] design each layer to be equivariant. One additional benefit of using equivariant models explained by our Theorem 4 and Remark 3 is that the optimization problem of score-matching using equivariant models on unaugmented data is equivalent to the optimization problem of score-matching using the same equivariant models on augmented data without manually doing data augmentation, so you still gain the term $\mathbb{E}[\mathbf{d}_1(\pi_G^N,\pi)]$ in Eq. (21) even if they are trained on unaugmented data, and the DFE is ensured to be 0.
> > >
> > > W2: We agree that Section 4.1 is not directly connected to the subsequent theories in Section 4.2 and Section 5. We will move it after the current Section 5 to be a separate section before the numerical section.
> > >
> > > W3: Following your comment in W2, we agree that it is better to move the MFG/HJB section to after the section on the significance of equivariant vector fields in SGMs. In the new section about the MFG, we will emphasize the assumption there is more general for any $G$. Also, we want to highlight again that HJB for any $G$ gives a separate justification that is more general. Thank you for this constructive comment!
> > >
> > >
> > > "It is an interesting point that indeed one can redefine the scalar product, such that the action is unitary - which seems interesting, but it is not immediately clear how the wasserstein-1 distance based results change. Do you have any intuition for this?" Given any compact group G and its non-unitary representation, we can derive a unitary representation of $G$ with a change of variables by the Peter–Weyl Theorem. Then we can apply our analysis (with the forward-backward processes) to the new coordinate system.
> > >
> > > W4: For example, in Group Equivariant Convolutional Networks (G-CNNs) [Cohen \& Welling, ICML 2016]: in the abstract, it says "G-CNNs use G-convolutions, a new type of
> > > layer that enjoys a substantially higher degree of
> > > weight sharing than regular convolution layers.
> > > G-convolutions increase the expressive capacity
> > > of the network without increasing the number of
> > > parameters. Group convolution layers are easy
> > > to use and can be implemented with negligible
> > > computational overhead for discrete groups generated by translations, reflections and rotations"; and before Secion 7.1, it says "The computational cost of the algorithm presented here is
> > > roughly equal to that of a planar convolution with a filter bank that is the same size as the augmented filter bank used
> > > in the G-convolution, because the cost of the filter transformation is negligible."
> > >
> > > Q2: Suppose $\rho$ is $G$-invariant, meaning $\rho(gx)=\rho(x)$. Taking the gradient with respect to $x$ on both sides, we have $A_g^\top\cdot\nabla\rho(gx)=\nabla\rho(x)$. Then left multiply both sides by $A_g$, we have $\nabla\rho(gx)=A_g\cdot\nabla\rho(x)$, which satisfies (3).
> > >
> > > Q4: Combining your suggestions in W2 and W3, we agree it would be better to separate the MFG section to a later section and clarify it holds for arbitrary groups, without assumption 1.
> > >
> > > Q5: We agree with the reviewer that it is more appropriate to move the symplectic integrator to the Introduction. Thank you again for the constructive feedback!
> > >
> > > "In addition to all of the above, the rest of the questions I have raised appear to be unanswered." -- Due to the limitation of the number of words in each response, we responded to your Q6-Q10 in a different block named "Author Response continued" right after the first part of our responses.

---

> > > > ### Comment · Reviewer_Y16Q · 2024-11-21
> > > >
> > > > Thank you for your quick response! I have indeed missed the second response and will engage with it in a separate comment.
> > > >
> > > > * W1: "using the reynolds averaging is not the only way to build equivariant models, for example, G-CNNs [Cohen & Welling, ICML 2016] design each layer to be equivariant." - this is precisely my problem with your claims. You claim that your analysis extends to such models, to say that G-CNNs would be 'better' than normal CNNs. However that is *not* what your results show. While it may be possible to show this by considering each layer separately (maybe) - your current theory shows this only for the reynolds operator.
> > > >
> > > > * Peter-Weyl theorem: "Given any compact group G and its non-unitary representation, we can derive a unitary representation of with a change of variables by the Peter–Weyl Theorem. Then we can apply our analysis (with the forward-backward processes) to the new coordinate system." - the action itself is fixed for whatever problem you are considering. Thus when you say new coordinate system you are referring to new coordinates on X. However, changing coordinates on X may change the metric (the new basis may not be normalized, especially if original action was not det Jac=1). Thus L1 on original space would not be L1 on the new space. Am I missing something in this line of thought?
> > > >
> > > > * W4: My original comment was not a questions of parameters - it was about difficulty of optimizing "this seems to completely miss the point of non-optimisability. Oftentimes, optimizing equivariant functions is harder/more expensive. And while errors can be made as small as possible in theory, practice will be different." And the papers provided do not seem to address this question at all.

---

> ### Author Response · Authors · 2024-11-20
> **Author Response continued**
>
> 6.  This is an important property of score-matching with equivariant vector fields, that you essentially gain data augmentation for free by simply using equivariant vector fields. Therefore, it is not clear to us why Remark 3 is *mathematically* obvious or known without the theorems from the objective function.
>
> 7. We want to clarify that we do not assume $e_{nn}=0$. In fact, quite the opposite, we rigorously show in Theorem 5 that the score approximation error has a lower bound, quantified by the DFE when using a non-equivariant score approximation (NN), even assuming perfect training and optimization. The comment of the reviewer ``optimizing equivariant functions is harder/more expensive'' is widely understood and addressed in the existing literature. In particular, while this is not the focus of our work, this statement contradicts many existing works related to equivariant neural networks, which are more efficient and easy to implement, such as G-CNNs. Also, the paper (Lu et al., 2024) cited in our paper provides numerous practical implementations.
>
> 8. The blue and orange curves are not expected to overlap: one represents results with *data augmentation* on the training samples, while the other does not, and both are plotted using 25 replicates. Since the samples are drawn randomly, some variation is naturally expected. As for the observation, “Even more surprisingly—why are the red and green curves so close?”, this outcome is entirely consistent with our quantitative discussion in Section 5, which builds on our theoretical results. Specifically, data augmentation offers limited benefits when the network is non-equivariant, as it cannot reduce the DFE error in such cases. This behavior is independent of data augmentation itself. Overall, Figure 1 supports our theoretical conclusion that leveraging equivariant score approximations is fundamentally more impactful than relying solely on data augmentation.
>
>
> 9. We want to correct the reviewer that our approximation of Wasserstein-1 distance $\mathbf{d}_1$ is different from WGANs. First, we do not train a generator. Second, we do not use mini-batches when computing the approximate $\mathbf{d}_1$ between generated and true samples. Moreover, we do not use gradient penalty but rather apply spectral normalization, which offers a type of ``neural net distance'' [R1] that approximates $\mathbf{d}_1$.
>
> [R1] Arora, S., Ge, R., Liang, Y., Ma, T. and Zhang, Y., 2017, July. Generalization and equilibrium in generative adversarial nets (GANs). In International conference on machine learning (pp. 224-232). PMLR.
>
> 10. Line 491: our simulation results are not extensive, but they corroborate our theory. Training an equivariant network on augmented data is part of our ablation study that makes the comparisons in Figure 1 complete. More empirical results can be found in (Lu et al., 2024) cited in our paper. Training equivariant NNs on augmented data makes sense for the completeness of the experiments. We also want to remind the reviewer that ``the parameter gradient are the same in both cases" is not true in general, as it also depends on the loss function. Your statement is actually supported by our Theorem 4.

---

> > ### Comment · Reviewer_Y16Q · 2024-11-21
> >
> > 6. Agreed - I see now that the point you are making is that it is equivariant for all times T - in which case it indeed is not an obvious claim. Am I interpreting this correctly now?
> >
> > 7. But e_{nn} depends directly on the parameterisation, correct? Thus I guess the most accurate claim for the paper would be that "Equivariant models with the same error in training would be better than their non-equivariant counterparts", and this would indeed be correct, unlike the current claim.
> >
> > I am also slightly confused - first you say that  ''optimizing equivariant functions is harder/more expensive is widely understood and addressed in the existing literature.'' but then you state that ''this statement contradicts many existing works'' - which is it? Either I am not correct in my statement (which may be the case), or it is correct - in which case this is something that needs to be mentioned in the paper.
> >
> > 8. Both blue and orange are equivariant models - therefore, for any augmented data-point, there is a non-augmented one which gives the same gradients in training. Thus, the only case in which they would not overlap - is due to stochasticity in batch selection. If that were the case - the error bars would at least intersect. However, they do not - meaning that there must be another systematic - my guess would be the wasserstein distance approximation being incorrect.
> >
> > In regards to the non-equivariant models - your theory does *not* analyze the non data-augmented case, right? So i am not sure how you claim that it makes sense? The non-augmented model should not be able to generate a quarter of all possible samples, thus the support of augmented is extremely different from the support of the non-augmented one - how could it make sense that the distances agree?
> >
> > 9. I never claimed that you were training a WGAN - I realise that there is no generator. However, the point still stands - what is even more worrying is that you dont use minibatches, implying that the network used to approximate the d1 is likely to get stuck in local minimas? I am not sure whether spectral normalization solves the issue in the paper i mentioned - I believe such an evaluation is still flawed.
> >
> > 10. "We also want to remind the reviewer that ``the parameter gradient are the same in both cases" is not true in general, as it also depends on the loss function." - could you please expand on this? In my understanding, in simple terms augmentation implies you go from $l(\theta,x_1) + ... + l(\theta,x_n)$ to $\sum_g l(\theta,g \cdot x_1) + ... + l(\theta,g \cdot x_n)$ - implying that for equivariant models the gradients would be the same?

---

> > > ### Author Response · Authors · 2024-11-21
> > >
> > > 9. "what is even more worrying is that you dont use minibatches, implying that the network used to approximate the d1 is likely to get stuck in local minimas" -- We do not think it makes sense to use minibatches to compute the $\mathbf{d}_1$ distance between two empirical measure, as in the variational form Eq. (10) the discriminator should be the optimal Lipschitz function with respect to the two complete sets of samples. We think the reviewer still confuses WGAN with our approximation of $\mathbf{d}_1$: the local minima of GAN comes from the two-player dynamic between the generator and the discriminator, while in our case it is purely optimizing the discriminator. So there is no such issue as the generator failing to improve and getting stuck in a local minimum. Though calculation may not provide the exact $\mathbf{d}_1$ distance, it is still an IPM metric as shown in [R1] to approximate the $\mathbf{d}_1$ distance, and we implemented it consistently as we fixed the NN architecture.
> > >
> > >
> > > 10. Following your notation, if $l(\theta,gx_i)\neq l(\theta,x_i)$, then the gradients would certainly be different. This means your loss function should respect or be invariant to the group action.

---

> ### Author Response · Authors · 2024-11-21
>
> We thank the reviewer for the quick response and appreciate the discussion. Please see our responses to your two sets of comments below.
>
> W1: We want to clarify that we do not claim "G-CNNs would be 'better' than normal CNNs" generically: for example, we should not expect a G-CNN with very simple architecture to perform better than a normal CNN with a much larger depth and number of channels and filters. What we claim through the logic of our theory is that: whenever you have a non-equivariant vector field, you should use its Reynolds averaging, meaning you should always use an equivariant vector field or NN. However, within the space of equivariant vector fields or the family of equivariant NNs, Reynolds averaging might not be the most efficient way to build the equivariant models, and we leave space for any design of equivariant NNs, e.g., G-CNNs are equivariant and are not designed by averaging. Besides, we also want to emphasize that the specific design and training of equivariant NNs are not the focus of our work. We think this will be a good clarification, and will add it to the revision.
>
> With a change of variables, we should define all the forward-backward processes and the score-matching objectives in the new coordinate systems, in which the matrix representation of $G$ is unitary. Then all our analysis can be applied to the new coordinate system. Finally, you can always reverse the transformation by the change of variables on the generated samples.
>
> W4: The training/expense of equivariant NNs is not the focus of our work and is a complementary task to our paper. However, we want to refer the reviewer to the empirical results in G-CNNs [Cohen \& Welling, ICML 2016]: for example, in Section 8.1 Rotated MNIST, they empirically compare the normal CNN with G-CNNs with approximately the same number of parameters and use the Adam algorithm for optimization, and the result illustrates that G-CNNs achieve smaller errors (almost halves the error) than normal CNNs.
>
> 6. Yes, it is for all time $T$.
>
> 7. "Equivariant models with the same error in training would be better than their non-equivariant counterparts" -- Your conclusion is correct regarding Eq. (21), but it is not the only possible conclusion. Another interpretation is what we explained in the response to point W1 above.
>
> We meant "optimizing equivariant functions is harder/more expensive" contradicts the empirical results in the existing literature. See also our response to W4. However, the training/expense of equivariant NNs is not the focus of our work.
>
> 8. "Thus, the only case in which they would not overlap - is due to stochasticity in batch selection. If that were the case - the error bars would at least intersect. However, they do not - meaning that there must be another systematic - my guess would be the wasserstein distance approximation being incorrect." -- The stochasticity comes also from random samples drawn from the target distribution. The *error bars* of the blue and orange curves (both using equivariant NNs) indeed intersect a lot of times. To be specific, the blue and orange curves are the bottom two curves in the range between $10^3$ and $10^4$ for $N$. On the other hand, the error bars of the red and green curves (both using non-equivariant NNs) also intersect a lot of times.
>
> "In regards to the non-equivariant models - your theory does not analyze the non data-augmented case, right?" -- Our analysis is built upon the non-equivariant case with no data augmentation (Mimikos-Stamatopoulos et al., 2024) cited in our paper.
>
> "So i am not sure how you claim that it makes sense? The non-augmented model should not be able to generate a quarter of all possible samples, thus the support of augmented is extremely different from the support of the non-augmented one - how could it make sense that the distances agree?" -- To clarify any misunderstanding about data augmentation, here is our explanation. In our experiment, the target distribution is a mixture of 4 Gaussians. Even with no data augmentation, you can still draw samples from the four centers and it is unlikely your samples are all clustered at one center. As a result, even use a non-equivariant NN, you can still generate a distribution from which you can observe four centers. Regarding the theorem, our Theorem 4 shows that the optimal equivariant vector field can be obtained by score-matching for raw training data without data augmentation whenever you apply equivariant NNs. For example in our Figure 2, in which we use 40 training samples, the probability of missing one quarter is $4\times0.75^{40}\approx 10^{-5}$.
>
> For responses to 9. and 10., see the next block.

---

> > ### Comment · Reviewer_Y16Q · 2024-11-23
> >
> > Fristly, I would like to thank the authors for engaging in the discussion and I am happy to say that I now feel more confident about contributions of this paper.
> >
> > W1,W4 and 7: I do agree that there may be more efficient ways to build equivariant models, however I would argue that this precisely is the problematic statement to make in generality. For simple groups, like those analysed emprically in [Cohen & Welling, ICML 2016], I fully agree. However efficiently constructing equivariant architectures with respect to general continuous groups is not an answered questions to the best of my knowledge. I fully agree that this is outside the scope of your current work and I do not expect you to answer this question in this work. I do, however, believe that this is something worth emphasising. To make this precise, the statements that you are trying to claim, that data augmentation is *worse* only holds under the assumption of having expressive parameterisation, and being able to optimize this parameterisations efficiently.
> >
> > * In the abstract you claim "Numerical simulations corroborate our analysis and highlight that data augmentations cannot replace the role of equivariant vector fields." - I do not belive that the statement that they "cannot replace" to be true. They *may not*, but only under assumptions stated above the *would not* be able to replace.
> >
> > 8: "The error bars of the blue and orange curves (both using equivariant NNs) indeed intersect a lot of times" they indeed do, but not for the first few data-points. I see now that this may be due to a very small number of samples - is this the reason why they do not?
> >
> > 9. Again, there is no confusion with GANs. I am talking about local minimas of the objective in the Kantorovich formulation $
> > \sup \left\{\mathbb{E}_\eta[\psi]-\mathbb{E}_\pi[\psi]: \psi \in \operatorname{Lip}_1(\Omega)\right\}$ - (or inf of - the objective). Even when optimizing over the full function space of lipschitz one functions I do not believe that the minimizer $\psi$ is unique. But even if it were, by invoking a parameterization this is no longer the case - there may exist multiple minimas in the parameter space. These are the minimas I am referring to - there is no reason to believe you actually find the supremum above, is there? What is [R1]?
> >
> > 10. The loss is usually a function of the network - and if the the network is indeed symmetric, $
> > l\left(\theta, g x_i\right) = l\left(\theta, x_i\right)$, no?

---

> > > ### Author Response · Authors · 2024-11-23
> > >
> > > We thank the reviewer for engaging in the discussion, which has also provided a few points for us to clarify our theory better in the revision. We are pleased that the reviewer has become more confident about the contributions of our paper.
> > >
> > > W1, W4 and 7: "However efficiently constructing equivariant architectures with respect to general continuous groups is not an answered questions to the best of my knowledge." -- Yes, to our knowledge the design of equivariant NNs for general continuous groups remains an open problem, and we will mention this point after the theorem and in our future work section. Also we want to emphasize that while it is possible to construct equivariant architectures with respect to general continuous groups, augmenting data over a continuous group is impractical.
> > >
> > > "To make this precise, the statements that you are trying to claim, that data augmentation is worse only holds under the assumption of having expressive parameterisation, and being able to optimize this parameterisations efficiently." -- We agree with the reviewer on this comment, our theory does not consider the specific optimization cost of equivariant NN and assumes the optimization is not a problem, which can be another important work. We also refer to the computational results in (Lu et al 2024) (see our references) for several implementation concepts for equivariant NN and related empirical findings.
> > > We will include these points in our current Section 5. Also, note that Reynolds averaging does not reduce the expressive power for equivariant vector fields. We have also mentioned "For instance, it would be valuable to explore the architecture of equivariant neural networks to ensure they possess sufficient expressive power while maintaining a
> > > manageable number of parameters," in our Conclusion and Future Work, and we will add the perspective of practical optimization.
> > >
> > > "I do not believe that the statement that they "cannot replace" to be true. They may not, but only under assumptions stated above the would not be able to replace." --  Let's consider the  two cases where the group is either finite or continuous. If the group is finite, then we can do data augmentation, also we can design equivariant NNs (at least you can use Reynolds average). Based on our theory, equivariant models produce smaller generalization errors due to the DFE. For infinite groups, we are sure we can not do a complete  and exact data augmentation. However, the latter issue could be resolved if equivariant NNs for continuous groups were available, though the problem is still open to our knowledge. Moreover, by our Theorem 4 (which is also true for continuous groups), once we have such architectures, the score-matching with non-augmented data automatically is equivalent to the optimization problem with augmented data, hence we could obtain data augmentation for free in this sense. We will add this discussion to our current Section 5, and we appreciate the discussion with the reviewer on these points to clarify our results and remaining challenges better.
> > >
> > > 8. Yes, when the number of samples is small, the third term $\mathbb{E}[\mathbf{d}_1(\pi_G^N,\pi)]$ dominates the right-hand side of Eq. (21), and has a large variance or randomness. Outside the small number of samples regime, the overall trend is consistent as $N$ grows.
> > >
> > > See our responses to 9 and 10 in the next block.

---

> > > ### Author Response · Authors · 2024-11-23
> > >
> > > 9. [R1] appeared in our early response, and we copy it below. We also want to refer to Theorem 8 and the  Examples on page 12 of [R2] that the set of ReLU neural networks with spectral normalization can be used to define probability metrics and divergences. Both [R1,R2] are published papers in a leading venue or journal, while https://arxiv.org/pdf/2103.01678 is an older preprint that has not been peer-reviewed. Also, we would like to mention that computing various Wasserstein distances almost always needs some sort of approximation (Sinkhorn, neural etc), while the same is also true in the vast literature of neural or RKHS estimators of divergences.
> > > While we appreciate the discussions on neural estimators, they are not the main focus of our paper. Relevant to our work, Figure 1 uses the sample NN architectures to compute the approximate $\mathbf{d}_1$ distance for samples generated by equivariant and non-equivariant models. It is clear which method produces a smaller error in the overall trend. In this sense, we do not think it is relevant that the distance is consistently underestimated or underestimated significantly more only for samples generated by equivariant models, given we always use the same discriminator architecture, the same neural estimator for Wasserstein-1 metric, the same number of generated samples, and the same fixed set of reference samples from the target for all cases in Figure 1.
> > >
> > > [R1] Arora, S., Ge, R., Liang, Y., Ma, T. and Zhang, Y., 2017, July. Generalization and equilibrium in generative adversarial nets (GANs). In International Conference on Machine Learning (pp. 224-232). PMLR.
> > >
> > > [R2] Birrell, J., Dupuis, P., Katsoulakis, M.A., Pantazis, Y. and Rey-Bellet, L., 2022. (f, Gamma)-Divergences: Interpolating between f-Divergences and Integral Probability Metrics. Journal of Machine Learning Research, 23(39), pp.1-70.
> > >
> > > 10. The loss function itself is not parametrized by the NN, so even though the NN is equivariant, the loss function may not be. Actually the $l(\theta,x_i)$ in your comment is the *network output*, not the loss function. It might be better to write $y_i = f(\theta,x_i)$, where $f$ is the NN function which is $G$-equivariant and $\theta$ is the NN parameter, such that $f(\theta,gx_i)=A_g y_i$. The empirical loss for non-augmented data is $L=\sum_i l(y_i)$, where the loss function $l$ may produce different values at $y_i$ and $A_gy_i$ if $l$ is not invariant to group actions. In our case, the loss function is the score-matching objective. Our Theorem 4 indeed supports your argument about the equivalent gradient flows of $\theta$.

---

### Author Response · Authors · 2024-11-20
**Global Response**

We thank the reviewers for unanimously acknowledging that our work 1)  provides the first theoretical guarantees of score-based generative models (SGMs) for learning distributions with symmetries, and 2) offers the first rigorous comparison between data augmentation and building symmetry into generative models.

We also want to emphasize that building symmetry into models is empirically more efficient than augmenting training data in e.g., structure-preserving GANs  (Birrell et al., 2022) and equivariant normalizing flows  (Kohler et al., 2020; Garcia Satorras et al., 2021), and equivariant diffusion models (Hoogeboom et al., 2022; Lu et al., 2024). They all provided empirical evidence that using equivariant/invariant models is more essential than data augmentation. However, no theoretical explanation for why model equivariance is more essential than data augmentation was provided in any of them.


The main suggestion by the reviewers is to include additional experiments; however, we emphasize again that the main purpose of the present work is its novel and substantial theoretical analyses. Moreover, we refer the reviewers to the recent paper (Lu et al., 2024), cited in our paper, for extensive experiments and implementations for equivariant SGMs, which also directly inspired our current *theoretical* study. However, our primary contribution is to *explain* the existing widely observed empirical results and to provide theoretical justification for the use of equivariant models in contrast to data augmentation.

---

### Author Response · Authors · 2024-11-27
**Revision updated**

We appreciate the feedback from the reviewers, and we have uploaded a revision. All the changes have been marked in blue color. Specifically,

1. We follow the discussion with reviewer Y16Q to move the HJB to a later section (now Section 6) before the numerical section. And we add more clarifications in Section 5 and Section 8. We also removed the symplectic integrators paragraph to save space.

2. We added the explanation of the notation '$\lesssim$' in the proof in the Appendix.

---

### Meta-Review · Area_Chair_RKyu · 2024-12-23

**Metareview:**

This paper presents a theoretical analysis of score-based generative models (SGMs) for learning distributions with symmetries. The authors claim to provide theoretical guarantees for SGMs in this context, offering generalization bounds for group-invariant data distributions. They use Hamilton-Jacobi-Bellman theory to describe the inductive bias of equivariant SGMs and argue that equivariant vector fields can learn symmetrized distributions without data augmentation.

The paper attempts to provide a theoretical framework for understanding symmetry in SGMs and offers mathematical analysis of equivariant structures in generative models. However, it suffers from limited numerical experiments, with only a simple one-dimensional example. The theoretical results may have limited practical applicability, and there is a lack of comprehensive comparison with existing methods in high-dimensional settings. The relevance of some theoretical results to the main claims remains unclear.

The paper's contribution appears incremental, primarily extending existing work on SGMs and equivariance. The limited numerical experiments do not convincingly demonstrate the advantages of the proposed approach in real-world scenarios. While the theoretical analysis is detailed, its practical implications and relevance to state-of-the-art SGMs are not sufficiently demonstrated. Consequently, the paper falls short of providing a substantial advancement in the field of symmetry-aware generative models, which is expected for acceptance at ICLR.

**Additional Comments On Reviewer Discussion:**

The discussion primarily focused on concerns about the practical relevance of the theoretical results and the limited scope of numerical experiments. The authors emphasized their focus on theoretical contributions and referred to existing literature for empirical validations. Reviewers questioned the generality of the claims, the connection between different sections of the paper, and the novelty of the findings in relation to existing literature. While the authors provided clarifications, some concerns remained unresolved.

The simplicity of the numerical example was initially criticized, but later acknowledged as consistent with the paper's theoretical focus. However, questions about the broader relevance and incremental nature of the work persisted throughout the discussion.

After considering these points, I concluded that despite offering some theoretical insights, the paper lacks the comprehensive impact and practical relevance expected for acceptance at ICLR. The limited experimental validation and unclear practical implications of the theoretical results were key factors in the decision to reject.

---

### Decision · Program_Chairs · 2025-01-22

Reject